# Horizontal Ice Flow Impacts the Firn Structure of Greenland's Percolation Zone

5  Rosemary Leone[1], Joel Harper[1], Toby Meierbachtol[1], and Neil Humphrey[2]

[1]Department of Geosciences, University of Montana, Missoula MT 59812
[2]Geology and Geophysics, University of Wyoming, Laramie WY 82071

**ABSTRACT**. One-dimensional simulations of firn evolution neglect horizontal advection from ice flow, which transports the firn column across climate gradients as it is buried by
accumulation. Using a suite of model runs, we demonstrate the impacts of horizontal advection on the development of firn density, temperature, and the stratigraphy of melt features through the Greenland ice sheet percolation zone. The simulations isolate processes in synthetic runs, and investigate four specific transects and an ice core site. Relative to one-dimensional simulations, the horizontal advection process tends to increase
the pore close-off depth, reduce the heat content, and decrease the frequency of melt features with depth by emplacing firn sourced from higher locations under increasingly warm and melt-affected surface conditions. Preservation of the advected pore space and cold content is strongly dependent upon the depth of meltwater infiltration. Horizontal ice flow interacts with topography, climate gradients, and meltwater infiltration to influence
the evolution of the firn column structure; the interaction between these variables modulates the impact of horizontal advection on firn at locations around Greenland. Pore close-off and firn temperature are mainly impacted in the lowermost 20-30 km of the percolation zone, which may be relevant to migration of the lower percolation zone. Relatively high in the percolation zone, however, the stratigraphy of melt features can have
an advection derived component that should not be conflated with changing climate.

## 1. INTRODUCTION

Summer melting of bare ice, epitomized by stream networks and moulins, represents a relatively small portion of the Greenland Ice Sheet (GrIS) periphery since about 90% of the ice sheet's area is perennially snow covered accumulation zone (e.g., Ettema et al., 2009). A large fraction of the snow covered region also experiences melt (Figure 1): between 50-80% of the area melted during summers of the period 1958-2009 (Fettweis et al., 2011), for example. Further, the inland extent and duration of melting have demonstrated increasing trends and have frequently established new records (Mote, 2007; Tedesco, 2007; Tedesco et al., 2013). Melting of the accumulation zone (i.e., the percolation zone) is therefore an increasingly important aspect of the ice sheet, and so too are the glaciological processes governing the snow/firn interactions with surface climate.

Meltwater from the lower accumulation zone may run off from its point of origin (e.g., Machguth et al., 2016), while at higher elevations the water may simply infiltrate into cold snow and firn to fill underlying pore space, forming ice when it refreezes (e.g., Braithwaite et al., 1994; Harper et al., 2012) or remaining liquid if it does not (e.g., Forster et al., 2014; Humphrey et al., 2012). The capacity of the firn column to accommodate meltwater is dependent on its thermal state, its density, and structure of ice layers. However, many aspects of the processes governing firn's structural and thermal evolution, and whether meltwater is retained, remain unclear. While current model fidelity prevents confident constraint on the amount of melt retained in the percolation zone, existing estimates are that 40-50% of the meltwater generated never escapes (van Angelen et al., 2013; Janssens & Huybrechts, 2000; Reijmer et al., 2012).

The percolation zone is a region with relatively high horizontal motion compared to submergence rate (*cf.* divide regions) (Figure 1). Ice sheet flow displaces the firn column to lower elevation, where it is buried by subsequent winter layers experiencing higher intensity summer melt. Thus, the deep firn column's structural makeup and thermal state results from a climate that varies in both time *and* space. Ice motion potentially impacts the structure of the firn column including the amount of deep pore space that could absorb meltwater and act as a source of associated latent heat. Further, it may have implications

for the interpretation of melt feature stratigraphy within ice cores collected from these regions.

Here we investigate the role that horizontal motion plays in driving the structural evolution of the firn layer in Greenland's percolation zone. We utilize previous approaches for modeling firn densification and meltwater infiltration, but we extend our analysis to include horizontal advection of the domain due to ice flow. Climate and ice flow are highly variable around the ice sheet, but sparse observational data, computational limitations, and questions of fidelity surrounding models for meltwater infiltration in firn, prevent us from simulating the entire ice sheet with a high level of confidence. Our purpose here is therefore to test for the importance of ice flow to firn structure in Greenland's percolation zone, because ice flow has largely been overlooked and is not currently included in regional climate model simulations of firn evolution. To explore this process, we focus our investigation on synthetic modeling of isolated processes, four differing transects of the GrIS percolation zone, and partitioning the signal of climate change from an advection signal within an ice core from the percolation zone.

**2. METHODS**

2.1 Model Description

Air temperature, accumulation rate, and melt/refreezing processes drive the evolution of the density and thermal structure of the percolation zone's firn column (e.g., Herron and Langway, 1980; Reeh et al., 2005). The spatial gradients in these parameters, coupled with the speed at which the ice moves through the gradients, also impacts evolution of the firn column. We simulate these processes in a thermo-mechanically coupled framework for firn densification and heat transfer, including meltwater penetration and refreezing, that also incorporates horizontal displacement of the firn column due to ice motion.

2.1.1 Firn Densification

The rate of change of firn densification is described by its material derivative:

$$\frac{D\rho}{Dt} = \frac{\partial\rho}{\partial t} + \mathbf{u} \cdot \nabla\rho \tag{1}$$

where $\rho$ is firn density and $\mathbf{u}$ is the velocity vector. Many existing densification models are based on the empirical assumption that the proportional density change is linearly related to the change in overlying load (Robin, 1958), and ours is no different. As rate parameters, we use the empirical constants developed by Herron and Langway (1980) for dry snow densification, based upon the relatively simplistic formulation with few tuning parameters and favorable comparison with other densification schemes (Lundin et al., 2017). The total densification rate is therefore:

$$\frac{D\rho}{Dt} = \begin{cases} c_0(\rho_i - \rho) & \text{if} \quad \rho \leq \rho_c \\ c_1(\rho_i - \rho) & \text{if} \quad \rho_c < \rho \end{cases} \tag{2}$$

where the critical density $\rho_c = 550 \text{ kg m}^{-3}$, and temperature and accumulation-dependent constants $c_0$ and $c_1$ are defined as:

$$\begin{cases} c_0 = 11\left(\frac{\rho_i}{\rho_w}\right) b \cdot \exp\left(\frac{-10160}{RT}\right) & \text{if} \quad \rho \leq \rho_c \\ c_1 = 575\left(\frac{\rho_i}{\rho_w} b\right)^{0.5} \cdot \exp\left(\frac{-21400}{RT}\right) & \text{if} \quad \rho_c < \rho. \end{cases} \tag{3}$$

In Equation 3, $R$ is the gas constant (8.314 J K$^{-1}$ mol$^{-1}$), accumulation rate $b$ is in ice equivalent units, and T is the absolute temperature.

2.1.2 Temperature Evolution

Firn temperature is simulated using the standard heat transfer equation, including latent heat additions from meltwater refreezing (see Section 2.1.3):

$$\rho c \frac{\partial T}{\partial t} = \nabla \cdot (K\nabla T) - \rho c\mathbf{u} \cdot \nabla T + S \tag{4}$$

where c is the specific heat capacity of ice, taken to be 2100 J kg$^{-1}$ K$^{-1}$, and S reflects latent heat release from refreezing. In Equation 4, K is the thermal conductivity, which we prescribe to be density-dependent following Arthern and Wingham (1998) ($K=2.1(\rho/\rho_i)^2$).

### 2.1.3 Melt Infiltration Schemes

Modeling complex and heterogeneous meltwater infiltration in firn with high fidelity remains an outstanding problem of critical importance and is beyond the scope of this project. Our approach is to implement three existing infiltration schemes which vary in complexity and reflect a range of approximations. The first model considers only shallow infiltration, assuming that all meltwater refreezes in the top annual layer (Reeh et al., 2005). The second implements a standard bucket method (Kuipers Munneke et al., 2014; Ligtenberg et al., 2018), allowing meltwater infiltration as far as permitted by thresholds for cold content and irreducible water content (the latter of which is defined following Coléou and Lesaffre (1998)). Meltwater percolates until reaching either a firn layer in which the available liquid water fails to exceed the layer's irreducible water content or the pore close off density; any remaining meltwater runs off instantaneously. The third infiltration model implements a continuum approach (Meyer and Hewitt, 2017), simulating the physics of water flow based on Darcy's Law, and treating both saturated and unsaturated conditions.

### 2.1.4 Numerical Methods

We implement horizontal advection in modeling exercises using an approach that is Lagrangian in the horizontal and Eulerian in the vertical. One-dimensional firn profiles are initiated with conditions characteristic of the dry snow zone, and are transported through the percolation zone along a flowline following a prescribed horizontal velocity. Because porous firn is confined to relatively shallow depths (cf. to the ice sheet thickness), vertical shear is negligible and the horizontal velocity can be assumed to be equivalent to the surface value. Horizontal motion is therefore simulated by translating spatially varying surface conditions (temperature and accumulation rate) to time-varying boundary conditions following the prescribed velocity, and Equations 1 and 4 remain restricted to the vertical dimension.

This approach implicitly captures the horizontal advection terms in Equations 1 and 4, but neglects any influence of along-flow velocity variations on firn structure. Firn densification rates may be influenced by mass changes arising from strain thinning/thickening, as well

as stress enhancement induced by horizontal deviatoric stresses (Alley and Bentley, 1988),
but are likely to be negligible around the GrIS percolation zone (Supplemental Material
1.1).

In addition, our modeling approach neglects horizontal heat diffusion. To assess the
consequences of this, we tested our Lagrangian approach against an explicit 2D model for
densification and heat transport including horizontal diffusion in an Eulerian framework.
Results from these tests yielded negligibly different results (Supplemental Material S1.2;
Figure S1). We therefore continue with our Lagrangian approach, which streamlines
computational efficiency and permits flexible implementation of various melt infiltration
schemes.

Changes in temperature, and density were coupled together along with vertical velocity
and solved using the finite element library FeniCS (Logg et al., 2012) with Galerkin's
method. Dirichlet boundaries for state variables temperature, density ($\rho_0$=360 kg m$^{-3}$), and
vertical velocity (following the accumulation rate as $w_{sfc} = -b \cdot \frac{\rho_i}{\rho_0}$) are imposed at the
model surface, and vertical gradients in these variables are set to 0 at the model base. The
model domain in all experiments is 80 m in the vertical; mass added to the model surface
from accumulation is balanced by mass exiting the domain at the model base. We use a
time step of one year in all simulations.

2.3 Model Experiments

The influence of horizontal advection on firn structure at depth is dependent on ice flow
speed and spatial gradients in climate forcings (temperature, melt, and accumulation). We
conducted an initial test of model sensitivity to each of these variables to understand, in
isolation, the influence of changes in these processes on firn structure. We then applied the
model to four flowline transects across GrIS' percolation zone representing a spectrum of
ice sheet and climate conditions. Finally, we performed a model test at a site in the upper
percolation zone to investigate the influence of horizontal advection on melt feature
interpretation.

2.3.1 Sensitivity Analysis

Synthetic sensitivity tests were performed over an 80 km model domain with spatially invariant surface slope, horizontal velocity, and accumulation rate. Melt rates were assumed to increase linearly from 0 at the inland boundary, to a specified fraction of accumulation at the lower model boundary. Horizontal velocity, accumulation rate, and

180 melt rate were varied around a base scenario of 100 m $a^{-1}$, 0.5 m $a^{-1}$ ice equivalent, and 85% of accumulation respectively. Baseline conditions were chosen to approximately match conditions along the EGIG transect. Horizontal velocities, accumulation rate, and total melt were then varied across ranges of values spanning the conditions that may occur in the GrIS percolation zone: velocity was varied from 0 - 500 m $a^{-1}$, accumulation from 0.1

185 - 1.0 m $a^{-1}$ ice equivalent, and melt from 0-85% of the accumulation rate. For each combination of horizontal velocity, accumulation rate, and melt forcing, we also imposed three different surface temperature gradients based on surface slopes of 0.3°, 0.6°, and 0.8° with a temperature lapse rate of -7.4 °C/km (Fausto et al., 2009). Sensitivity test results were compared against 1D results at the end of the 80 km model domain. Further details of

190 the metric of comparison are presented in the Supplemental Material (section S2).

2.3.2 Greenland Transects

Our modeling approach was implemented at four test transects (Figure 1) around the GrIS percolation zone: 1) the well-studied EGIG transect in western GrIS, 2) a transect feeding

Jakobshavn Isbrae, 3) the K-transect in southwest GrIS, and 4) a transect extending into Helheim Glacier. These four study profiles were selected to capture a wide variety of ice sheet conditions (Table 1). The inland extent of the percolation zone was selected to correspond with the location where melt, as a fraction of accumulation rate, is equal to the value at Crawford Point (Figure 1); a choice based on temperature measurements

indicating melt infiltration at Crawford Point is insufficient to warm the firn above the annual average temperature (Humphrey et al., 2012). The initial condition for the interior boundary of each transect was the steady state profile based on the values given in Table 1.

Surface velocities along study transects were defined from satellite velocity data (Joughin et al., 2010), and RACMO2.3p2 (Noël et al., 2018) was used to select 1980-2016 average climate variables (Figure S3). This time period captures the increase in GrIS melt since the late 20th century (Fettweis et al., 2011). In addition to the transect simulations incorporating horizontal ice flow, in each transect we also completed 1D simulations at 600-1700 locations spaced at annual displacements (calculated from surface velocities) along the flowline. The latter were used for baseline comparisons of the effects of including or neglecting horizontal advection of the firn column.

### 2.3.3 Impact on Core Stratigraphy

A commonly used metric for quantifying changing climate conditions from firn cores is the annual increment of surface melt, or Melt Feature Percent (MFP) (Graeter et al., 2018; Kameda et al., 1995; Koerner, 1977; Trusel et al., 2018). To investigate the role that horizontal advection can play in MFP records from the percolation zone, we simulated the conditions leading to Crawford Point (69.877°N, 47.0102°W, 1997 m elev), located along the EGIG line. This site is relatively high in the percolation zone; in recent decades the average summer at this site experiences about 15 days of melt (Mote, 2007). Multiple shallow cores have been collected for density and temperature measurements (Harper et al., 2012; Humphrey et al., 2012), and in 2007 and a deep core was collected and interpreted within the context of GrIS melt history (Higgins, 2012). The site therefore offers an opportunity to assess the role of horizontal advection on interpretation of melt history in a core profiling the full firn column of the percolation zone.

We modeled the 2D firn evolution along a flow line beginning 22 km inland, and ending at Crawford Point using datasets for the modern state. This transect extent is chosen to ensure that the simulated conditions at Crawford Point contain no remnants of the initial condition. Ice surface geometry (Morlighem et al., 2017) and, mean (1980-2016) melt and snowfall values from RACMO2.3p2 (Noël et al., 2018) were used to determine spatial climate gradients. As with the other Greenland transects (Section 2.3.2), horizontal advection during burial is represented by present-day velocity datasets (Joughin et al., 2010). We employ the (Reeh et al., 2005) model for infiltration to be consistent with past

MFP observational studies which assume all annual melt is confined to the corresponding annual layer (e.g., Graeter et al., 2018; Kameda et al., 1995; Trusel et al., 2018). We assume the spatial gradients in input datasets have not changed over the century time scale required to simulate firn conditions at the bottom of the firn column. The validity of this assumption is unknown and perhaps tenuous; our intention, however, is a demonstration of the horizontal advection process constrained by ice sheet conditions. Furthermore, if there are in fact large time changes in gradients, this only increases complexity in the signal created by horizontal advection.

**3. RESULTS**

3.1 Sensitivity Tests

Including 2D horizontal advection in simulations of the percolation zone yields greater air content in the firn column and increased depth to pore close off compared to 1D results (Figure 2; Figure S2). Surface speeds approaching the upper limit of what may be expected in GrIS' percolation zone generate a firn column with air content that can differ from 1D simulations by 80%. Yet, while greater ice flow speed clearly influences horizontal advection-based results, the impacts are strongly modulated by the magnitudes and gradients in other variables. For example, the impact of horizontal advection is also a function of accumulation, with smaller accumulations causing a 25-35% increase in the depth to pore close off and total air content in 2D simulations relative to the 1D model runs. This stems from reduced densification rate under smaller annual increments of overburden, and thus longer preservation of cold and porous firn that becomes deeply buried firn further down-glacier. Adding melt gradients to the scenarios exacerbates the effect, with wet surface conditions overlying dryer conditions at depth.

In addition to surface climate variables, changes to the density structure by including horizontal advection are sensitive to the choice of meltwater infiltration scheme. The largest changes to density (and therefore air content) in the presence of advection occur in the Reeh et al. (2005) infiltration scheme. In contrast, firn air content results from the bucket scheme and continuum model are far less sensitive to advection (Figure 2). The

265 changes to air content from advection in these infiltration scheme are approximately half of the Reeh et al. (2005) across all tested velocities.

Adding horizontal advection to simulations also decreases the firn temperature; the temperature profile and temperature at pore close off reflect advected firn from higher,

colder conditions. Heat content is strongly influenced by the choice of meltwater routing scheme: for example, under very high accumulation and melt, the bucket method yields deep penetration of water and warmer firn temperature at depth (cf. the 1D case) (Figure S2). Steeper topography yields larger along-flow gradients in melt, temperature, and accumulation, causing greater disparities between 2D-advection and 1D-profile

simulations. The ice flow speed has potential to impact simulations with 2D-advection, but importantly, the magnitude is strongly modulated by the values and gradients in other variables. Interestingly, while meltwater infiltration and refreezing amplifies the difference between 1D and 2D-advection simulations of firn density, the release of latent heat eliminates advected cold content, thereby reducing the thermal disparities between 1D and

2D simulations (Figure S2).

3.2 Transects

The most significant differences between the 1D and 2D model simulations are along the lowermost 20-30 km of our four sample transects. By including ice flow in these firn

simulations, the density decreases by >50 kg m$^{-3}$ for the EGIG, Jakobshavn, and Helheim transects (Figure 3; Figure S4; Figure S5), resulting in increases of pore close off depth of up to 8 m, 13 m, and 19 m, respectively. The commensurate impacts on total air content in the firn column can also be large: for example, along the EGIG transect it changes by ~50% in the lower 10 km, and by 7%-25% along the next 10-20 km.

The different melt infiltration schemes yield variable impacts. As in the sensitivity testing, the largest impact is with the Reeh et al. (2005) scheme, under which the inclusion of horizontal advection in simulations increases the firn column air content by up to several meters from a 1D simulation (Figure 4). In contrast, the air content changes are almost

always less than one meter in the continuum scheme, which allows for deep meltwater penetration.

Local changes in surface slope along the transects both enhance and diminish the impacts of horizontal advection on the underlying firn structure, complicating the 2D geometry of the firn layer within the percolation zone. This is exemplified in both EGIG and Jakobshavn transects (Figure 3). A flattening of the surface profile near 20 km (EGIG) and 30 km (Jakobshavn) means that snow accumulates under similar climate conditions over many km. This generates a deep firn density structure that is more similar to 1D profiles as the firn is buried and horizontally advected through the transect. These flat regions terminate abruptly with a sharp increase in slope, and therefore climate gradients as well. This results in deep firn with densities that are reduced by 20-30 kg m$^{-3}$ compared to 1D simulations. In contrast to the EGIG and Jakobshavn transects, the changes to density structure throughout the K-transect are comparatively small. Surface speeds are consistently ~18 - 50% of the values along EGIG and Jakobshavn counterparts (Table 1); such slow velocity all but eliminates the impact of ice flow (Figure 3d).

The process of horizontal advection generates colder firn temperature profiles. Along the EGIG transect, horizontal advection decreases firn temperatures at the depth to pore close off by 1.0°-1.5° C in the lower 15 km, and by 0.8°-1.0° C in the next 20 km. With the high speeds, steep topography, and heavy melt of the lowermost reaches of Jakobshavn and Helheim transects, firn temperatures were reduced by as much as 3° C by including horizontal advection.

3.3. Impact on Core Stratigraphy

At Crawford Point, firn deposited along the first ~5km above Crawford Point reflects a region with very low slope and essentially no climate gradient caused by elevation (Figure 5A). Consequently, MFP values are relatively constant in the upper ~50m of the firn column, indicating that horizontal advection is inconsequential to firn structure over depths that are equivalent to recent decades at this site (Figure 5B).

Below ~50m depth, however, there is an abrupt inflection to continuously decreasing MFP to the bottom of the core (Figure 5B). Horizontal advection generates a decline in MFP of approximately 7 percentage points from 50 m depth to the end of the core. Presented in terms of simulated age (rather than depth), this amounts to an apparent reduction in melting of 0.04 percentage points per year that arises from horizontal ice flow and not time changes in climate. As discussed below, this is a non-trivial magnitude when scaled against changes in melt arising from warming climate.

**4. DISCUSSION**

4.1. Uncertainty due to Infiltration

The choice of meltwater infiltration scheme has a large effect on the simulated impacts of horizontal advection of firn in the percolation zone and is a key uncertainty in the fidelity of model results. In reality, water moves vertically as a wetting front propagating downward from the surface (Colbeck, 1975), but also by complex and inhomogeneous infiltration processes (Marsh and Woo, 1984; Pfeffer and Humphrey, 1996), and it can be routed horizontally along impermeable ice layers (e.g., Machguth et al., 2016). With so little known about deep infiltration, none of our schemes are likely to be entirely accurate: the Reeh et al. (2005) scheme only allows melt penetration within the annual snow increment which is known to be incorrect, especially low in the percolation zone where melt rates are high (e.g., Humphrey et al., 2012); the continuum model (Meyer and Hewitt, 2017) uses the most complex physics, but has large uncertainties for coefficients of permeability and grain sizes; and, the bucket model (Kuipers Munneke et al., 2014; Ligtenberg et al., 2018) disregards the complex physics governing flow of water through the firn matrix, simplifying the problem to just density and cold content and assuming the flow of meltwater is instantaneous.

With horizontal advection of firn tending to move open pore space underneath an increasingly melting surface, the depth/quantity of infiltration is key: the deeper melt penetrates, the more the deep pore space is 'overprinted' by surface melt and not preserved. Alternatively, infiltration that is limited to shallow depths enhances the disparity between deep firn and that nearer to the surface. Our suite of model runs show

that, in the lower percolation zone, the choice of infiltration scheme has nearly equivalent impact on the total air content as the incorporation of ice flow.

4.2 Spatial Variability of Firn Structure

Simulation results demonstrate the changing influence of horizontal advection on firn structure in response to changing climate gradients and ice speed as the firn parcel traverses the percolation zone. Transect modeling indicates that along any given flowline the influence of advection tends to increase towards the lower percolation zone; an

intuitive result considering that surface speed and slope (a proxy for climate gradients) both increase substantially relative to the upper percolation zone, and the surface experiences heavy melt (Supplemental Figure S3). Sensitivity testing showed that each of these factors amplifies differences between 2D and 1D representations of deep firn structure.

Transect modeling also reveals that ice flow introduces variability in firn structure around the ice sheet, not just along individual flow lines. Surface speeds within the GrIS' percolation zone shown in Figure 1 vary from nearly 0 to more than 1000 m a$^{-1}$. The K-Transect, EGIG, and Jakobshavn transects demonstrate the differences in firn structure that

can develop regionally across Western Greenland as a result of ice flow patterns (Figure 3). However, the EGIG and Jakobshavn simulations show that advection can also influence firn structure at a more local scale. Despite being separated by just ~40 km, differences in surface speed develop between the two transects in the lower 20-30 km as ice in the Jakobshavn transect accelerates towards the margin (Figure S3). This results in a simulated

firn column that is 5-10 m thicker compared to the nearby EGIG profile at the same elevation. While the local gradients in ice speed are perhaps greater here than nearly anywhere else on the ice sheet, the local and regional differences in our simulated transects illustrate that differences in deep firn structure are likely to exist in regions of the GrIS percolation zone with otherwise similar climate conditions, purely as a result of differences

in ice flow patterns and topography that dictate spatial gradients in climate.

4.3. Melt Feature Stratigraphy

A 152 m long ice core collected at Crawford Point in 2007 (Higgins, 2012; Porter and
Mosley-Thompson, 2014) offers the opportunity to compare measured data against our
modeled depth change in MFP stemming from horizontal advection. The core age extends
back to the year 1765 based on seasonal isotope variations, and the modeled flow field
shows the bottom of the core originated ~260 years prior and about ~22 km up the flow
line (Figure 5a). Thus, the flow model age estimate at the core-bottom is within 7% of the
age determined by isotope methods. Higgins (2012) measured an overall trend of
increasing MFP from 1765-2007 of 0.08 percentage points per year. However, melt events
prior to 1900 were minor and infrequent; the more recent trend from 1900-2007 therefore
increased to 0.11 percentage points per year.

The horizontal advection signal is also highly dependent on the defined time period, but for
another reason: different time periods sample different spatial gradients of the ice sheet
surface as firn moves across the percolation zone. At Crawford Point, the MFP signal in firn
from recent decades has almost no influence from horizontal advection because this firn
has formed along a local flat region in the topography extending about 5 km up flow from
the site (Figure 5a). Further inland, surface elevation (and therefore climate) gradients
increase. Older firn sourced from these regions displays less melt and lower MFP.

Over period 1765-2007, a MFP trend of 0.08 percentage points per year  was observed in
the core (Higgins, 2012) compared to 0.03 percentage points per year in our model that
includes ice flow but no warming. This indicates that more than one third of the MFP trend
over the entire core can be attributed to advection (Figure 5b). However, focusing on the
more recent period of 1900-2007, our simulation indicates a reduction in the advection-
controlled MFP trend to 0.02 percentage points per year. The decreased influence of
advection on the MFP trend in this recent interval, coupled with the increasing MFP events
measured in the core, support the conclusion that climate has warmed during this period.
Nevertheless, our results illustrate that the stratigraphy of melt features along an ice core
from the percolation zone can have a complex spatial component that should be evaluated
to properly interpret temporal change.

While the MFP trend at Crawford Point is impacted by horizontal advection, simulated profiles of firn density and temperature have little influence from the advection process relative to lower on the transect (Figure 3). This result may seem counterintuitive. However, the density and temperature fields evolve over a time-space continuum, whereas the MFP record represents a small time-trend in the occurrence of discrete events.

Occasional thin ice layers, which are easily identified in cores, can have little impact on the bulk density of the core. Further, the magnitude of trends sets the importance of horizontal advection in a MFP record. In the Crawford case, the multi-decadal trend in MFP due to changing melt is a fraction of a percentage point per year, an important indicator of changing climate, but not large enough to completely mask the signal from horizontal

advection. Thus, because important climate change signals can be on par with the advection signal, the process should be considered in interpretations of MFP records.

Certainly some locations in the percolation zone may yield ice cores with MFP trends that are not significantly impacted by ice flow. But considering the potential for ice flow to

435 obscure climate trends, a simple procedure for quantifying this effect has utility. If the present ice sheet state (speed, accumulation, and melt rates) is assumed to be constant in time, an apparent climate signal at any core site can be quantified from spatially extensive datasets of the above variables. At a dated core depth corresponding to a time $t$ years before present, the firn parcel originated at a location ($x(t)$) upglacier from the core

location, where $x(t)$ is the integral of the spatially varying velocity ($v$) along the flowline over $t$ years:

$$x(t) = -\int_0^t v(x) \, dt. \tag{5}$$

The MFP at time ($t$) can therefore be determined from the accumulation and melt conditions at this upglacier location:

$$MFP(x) = \frac{m(x)}{b(x)}. \tag{6}$$

Equations 5 and 6 can thus be combined to generate a time series of MFP at a core site that
is a record of spatially varying climate advected by ice flow; the component that should not
be incorrectly interpreted as time-changing climate.

## 5. CONCLUSIONS

Elevated horizontal ice flow in the percolation zone compared to ice divides results in a firn
column that is not always well represented by 1D models for time-evolving density and
temperature. The impacts of horizontal advection are highly variable around the ice sheet,
but accounting for horizontal advection in simulations can change the firn's air content by
10s of percent and the temperature can differ by several degrees. Lower accumulation,
higher velocity, higher melt, and steeper topography (which drives climate gradients) all
increase the mismatch between surface and deep conditions, exacerbating the
shortcomings of a 1D simulation. The horizontal advection process thus has greatest
influence on firn evolution in the lower accumulation zone (e.g., 20-30 km); a nexus of
conditions that are likely migrating upward as climate warms, but are also subject to the
greatest uncertainty regarding melt infiltration processes.

The 2D evolution of firn in the percolation zone is influenced by topography: horizontally
invariant firn is generated in flat regions, whereas local hills/swales enhance the 2D
influences from horizontal advection. The deeper meltwater penetrates, the more pore
space is filled by surface melt and the advected deep pore space and cold content is not
preserved. The stratigraphy of melt features along an ice core from the percolation zone
can have a strong spatially derived component. Melt feature stratigraphy can be impacted
by horizontal advection high in the percolation zone, where firn density and temperature
are relatively unaffected by ice flow. This effect must be evaluated to properly interpret
temporal changes in ice cores related to climate, especially over decadal and longer time
scales.

***Code/Data availability.*** This paper has no data to declare. All model simulations are available at https://github.com/um-qssi/Firn_advection.

***Author contributions.*** RL, JH and TM designed the approach, which RL then carried out. RL prepared the manuscript with contributions from all co-authors.

***Competing interests.*** The authors declare that they have no conflict of interest.

***Acknowledgements.*** Funded by NSF grants 1717241 (Harper, Meierbachtol) and 1717939
(Humphrey), and a Montana NASA Space Grant Fellowship (Leone). The authors thank J. Johnson for discussions, and B. Vandecrux and an anonymous reviewer, whose careful reviews improved the manuscript.

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

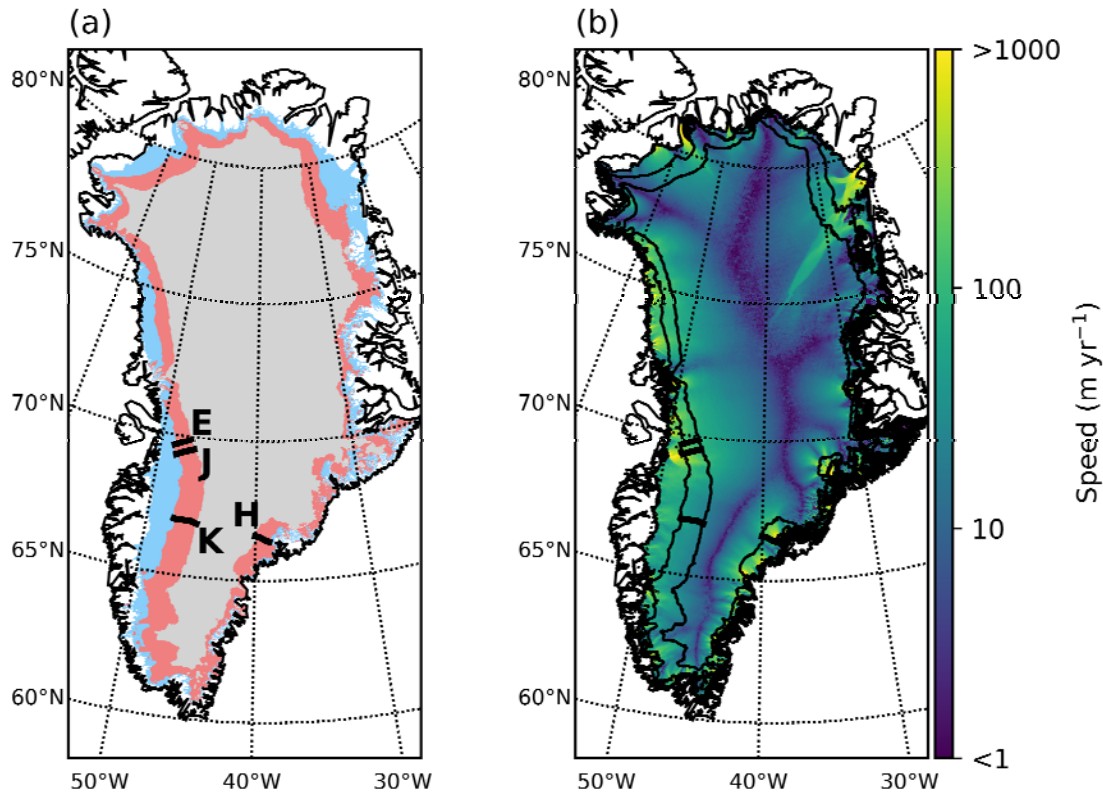

**Figure 1.** Maps of Greenland. (a) Facies delineated based on modeled 1980-2016 average surface melt (Noël et al., 2018): ablation zone (blue) with melt exceeding accumulation; percolation zone (red), the upper limit of which defined by melt conditions at Crawford Point where infiltration has not warmed firn (Humphrey et al., 2012); dry zone (gray). (b) velocity field from Joughin et al. (2010) with top and bottom of percolation zone shown in (a) delineated by black contour lines. Thick black lines through percolation zone show study transects, where E is EGIG, J is Jakobshavn, K is K-transect, H is Helheim (see Table 1).

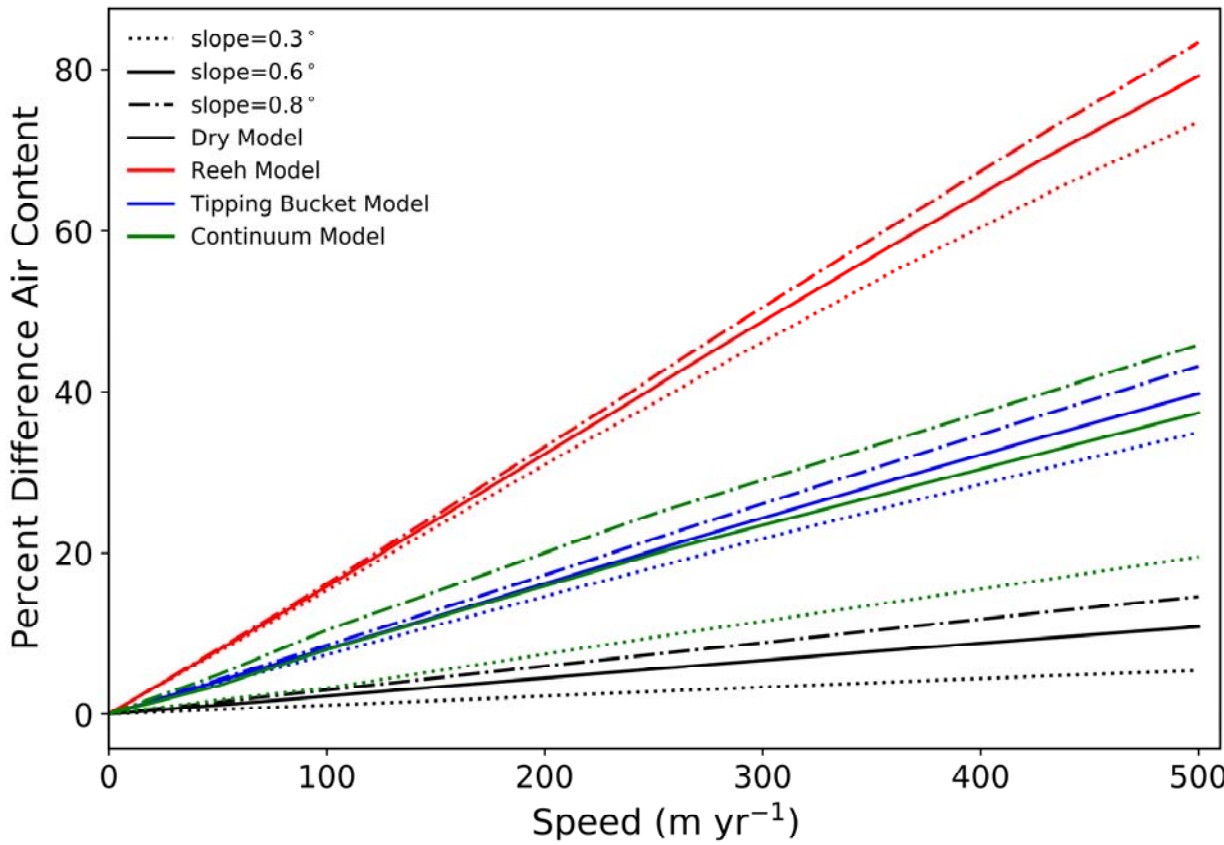

**Figure 2.** Example sensitivity test. Modeled differences between 1D and 2D for ice speed using dry firn model (black), Reeh model (red), tipping bucket model (blue), and continuum model (green). Base accumulation is 0.5 m a$^{-1}$, approximately the average value of the EGIG transect shown in Figure 1. Each simulation is run with surface slopes of 0.3°

(dotted), 0.6° (solid), and 0.8° (dash-dotted) to represent different climate gradients that may exist around the GrIS.

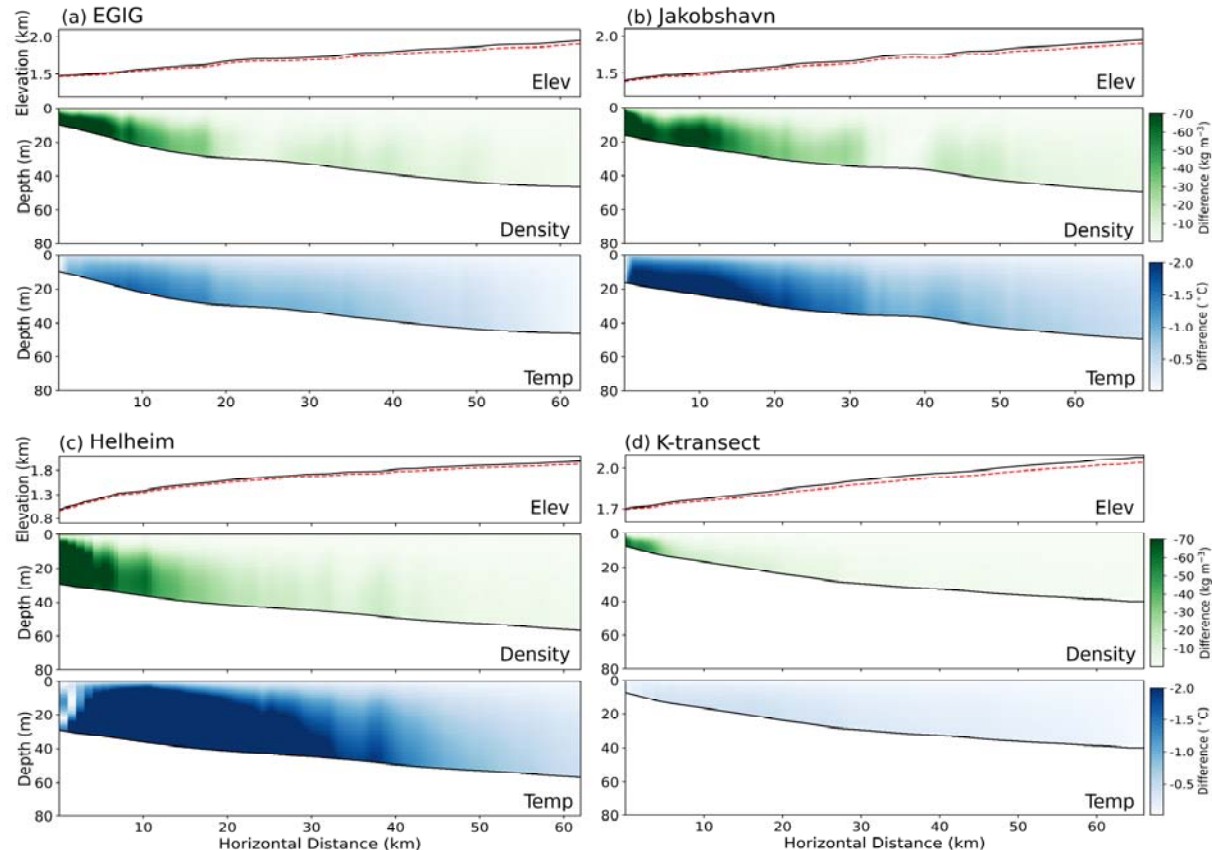

**Figure 3.** Calculated difference between 2D and 1D simulated firn properties in the percolation zone through the four study transects with bucket method meltwater infiltration scheme: a) EGIG line; b) Jakobshavn; c) Helheim; and, d) K-transect. Top panel in each transect shows surface topography (black) and pore close-off depth (red dashed). Middle panel shows density differences (2D - 1D), and bottom panel shows temperature differences. Different transect lengths reflect different percolation zone extents based on the definition stated in the text.

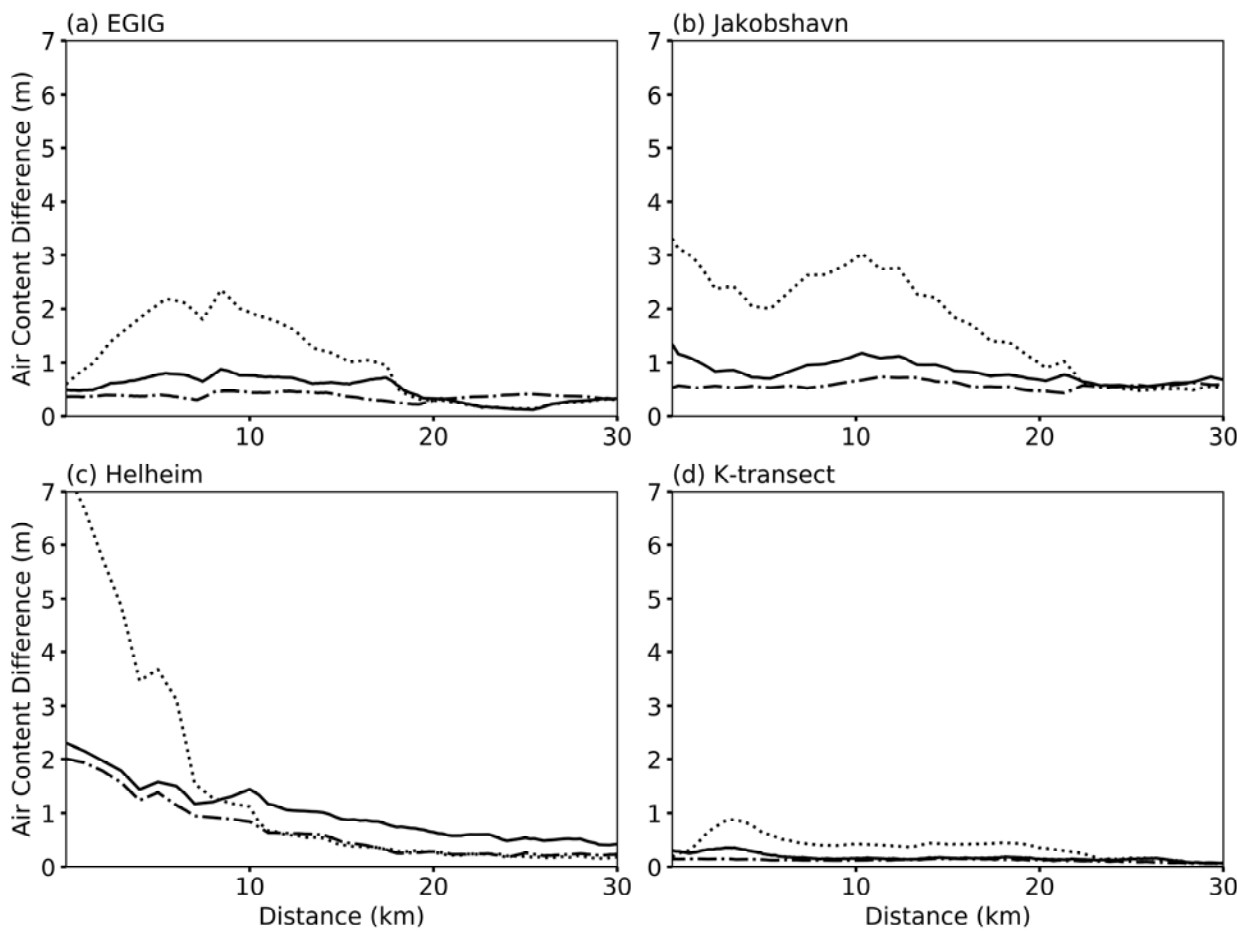

**Figure 4.** Simulated difference between in integrated firn air content 2D and 1D modeling schemes in the lowest 30 km of EGIG (a), Jakobshavn (b), Helheim (c), and K-transect (d). Differences are presented for each meltwater infiltration scheme: Reeh et al. (2005) (dotted), bucket method (solid), and continuum (dash-dotted).

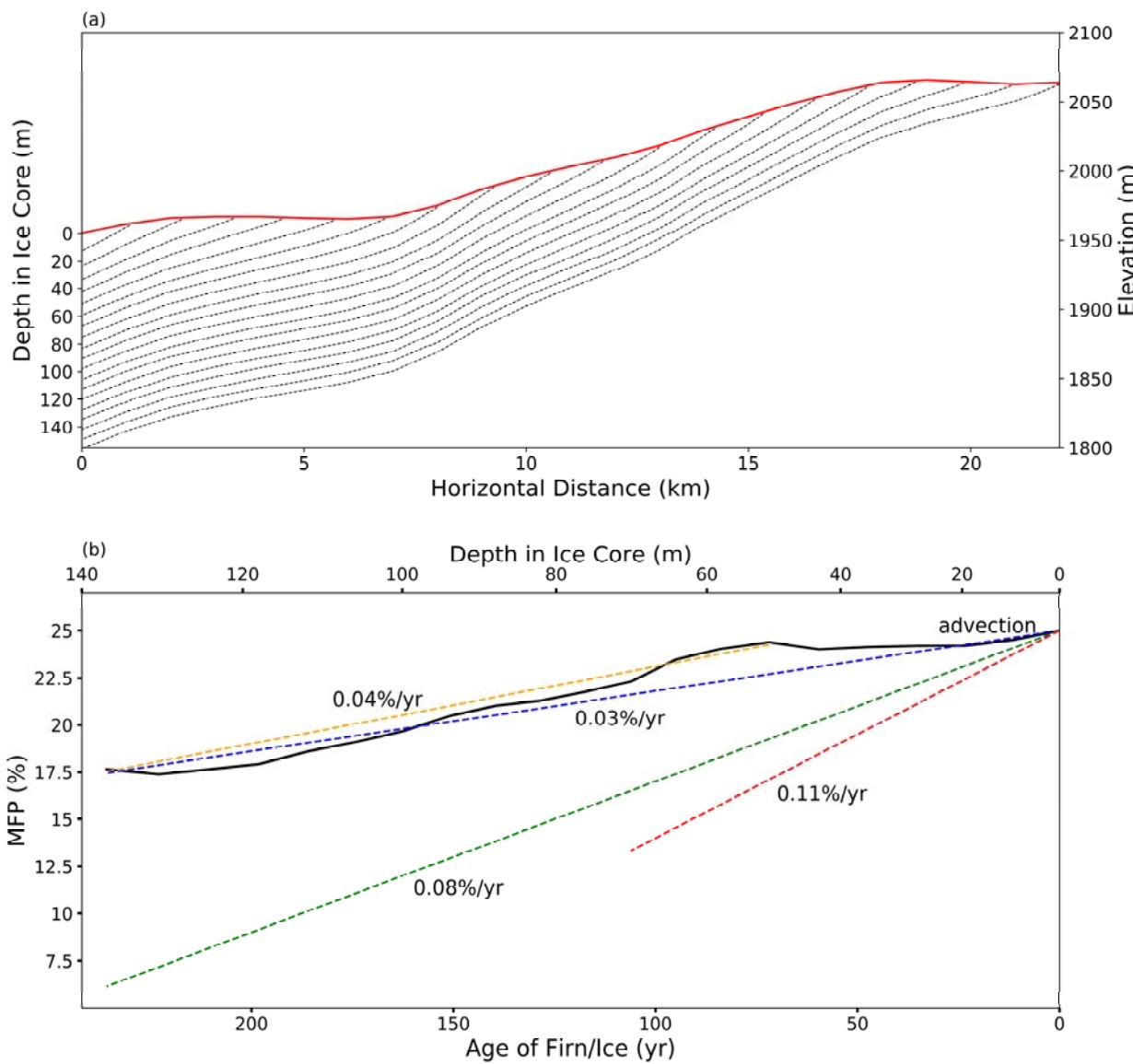

**Figure 5.** Surface topography and modeled flow lines extending inland from Crawford
Point (a). Horizontal distance scale is kilometers from Crawford Point. Bottom panel (b)
shows the modeled change in MFP over time (bottom axis) and with depth (top axis)
resulting from ice flow alone. Depth scale in (b) corresponds to firn depth in (a). Time
trends in MFP at Crawford Point arising from simulated ice flow are shown for the full
time/depth period (blue) and for the firn profile below 50 m (orange). Time trends in MFP
measured in a Crawford Point ice core and reported by Higgins (2012) entire period
(green) and the 1900-2007 period (red) are shown for reference.

**Table 1.** Approximate conditions taken from RACMO2.3p2 (Noël et al., 2018) along the four transects used in the study.

| Transect | EGIG | Jakobshavn | K-transect | Helheim |
|---|---|---|---|---|
| Elevation Range (m) | 1470-1950 | 1290-2020 | 1700-2082 | 1232-2160 |
| Speed (m yr$^{-1}$) | 93-150 | 85-400 | 27-71 | 35-1900 |
| Snowfall (m ice eq.) | 0.46 | 0.55 | 0.4 | 0.7-1.3 |
| Temperature (ºC) | -14º to -18º | -13º to -18º | -9º to -18º | -15º to -17º |
| Melt (m ice eq.) | 0.11-0.43 | 0.1-0.53 | 0.15-0.4 | 0.1-1.3 |