# Peer review of "Horizontal Ice Flow Impacts the Firn Structure of Greenland's Percolation Zone"

_The Cryosphere, 2019_

## Referee Comment (RC1) · Anonymous Referee #1 · 19 Nov 2019

**Review: "Advection Impacts the Firn Structure of Greenland's Percolation Zone"**

by Leone et al.

Submitted to *The Cryosphere*

**1 General**

In this paper, the authors study the role of downstream ice advection on firn compaction and meltwater percolation. They find that including ice advection leads to larger pore close-off depths, lower heat content within the firn, and a lower frequency of melt features as compared to one-dimensional simulations. What I like about this paper is that it (1) connects firn physics modeling to ice flow, which is novel and important, and (2) adds a new dimension to the firn compaction problem, where ice advection causes features at a given point on the ice sheet to inherit the upstream history rather than start from a blank slate, as is often assumed in the current paradigm. Astoundingly, the authors show that melt features can have an advection-derived component that is not due to climate change but rather incorrect model initial conditions. There is a lot of very interesting science in this paper and I would certainly like to see it published. Unfortunately, however, this paper requires major revisions before I would recommend publication in *The Cryosphere*.

**2 Remarks**

The overarching concern I have is the presentation of the model results. The methodology is poorly explained (e.g. how does FEniCS fit in?), meaning that it would be difficult to reproduce or verify the results. Moreover, the results that are shown are not very insightful as presented. I suggest: (1) clarifying in a general way how the models are queried and (2) making the figures more accessible. I give some suggestions for how to address these two comments below.

**3 Specific comments**

1. line 13: There are many missing hyphens between words, such as 'one dimensional' should be 'one-dimensional'.

2. line 14: I suggest 'compaction' instead of 'burial'.

3. line 38: I think that it is worth stating that these percentages are area and not volume.

4. line 68: the two-dimensional extension deserves much more explanation!

5. line 71: I suggest adding a paragraph describing the road map for the paper.

6. line 77: could add 'as well as compaction and advection', given that this is the topic of the paper and these clearly influence the density and thermal structure. This would strengthen the topic sentence.

7. line 84: 'rather' would be a good addition after the comma.

8. line 91: It makes sense that the one-dimension advection does a good job approximating the two-dimensional solution because the downstream gradients are small (viz. Hewitt and Schoof, 2017). The important issue, however, is that the one-dimensional advection *does not* include downstream transport of moisture, which is likely to be very important and is not included in the two-dimensional implementation described in the supplement, correct?

9. line 97: this paragraph is confusing as it makes it sound like the authors implemented everything in FEniCS — is this true?

10. line 120: why wasn't the Community Firn Model (Stevens, 2018) used? It contains several implementations of meltwater percolation through firn and is open source (https://github.com/UWGlaciology/CommunityFirnModel).

11. section 2.3: I think it would be very valuable to write to out the equations in a general way, so that it is clear (1) what advance the authors have made and (2) how the advance is implemented operationally. In other words, I suggest writing the density $\rho$ evolution equation as

$$\frac{d\rho}{dt} = f\left(\rho, T, ...\right), \tag{1}$$

where the right-hand side is given by Reeh et al. (2005) etc. is a function of the temperature $T$. Then, the authors could state that they will add downstream ice advection $u$ as

$$\frac{\partial \rho}{\partial t} + u\frac{\partial \rho}{\partial x} = f\left(\rho, T, ...\right), \tag{2}$$

and state that this advection process occurs explicitly (my assumption), where the upstream density is advected downstream at each timestep.

12. section 2.3.1: I suggest including some of the figures from the supplement in the main text to demonstrate how the models work.

13. line 276: 'for much another reason' should be 'for another reason'.

14. line 282: typo as there should be parentheses around the figure reference, i.e. '(Figure 5b)'.

15. line 302: 'x' should be '$x$'.

16. line 335: simulations produce data. How do I access the simulations? Also, I suggest putting Leone in parentheses to match the other funding acknowledgments.

17. line 378: error in title

18. figure 1: is this figure useful/insightful?

19. figure 2: I suggest addressing the dot, solid, and dot-dash lines within the caption. Also, an additional figure showing the two-dimensional results for the different models shown in figure 2 could be very useful (i.e. figure 3 but for the figure 2 simulations).

20. figure 3: it would be helpful to label the subfigures on the actual figure. For example, the transect name could be put next to the letter, i.e. '(a) EGIG line' and topography, pore close-off depth, and differences could be labeled on the respective panels.

21. figure 4: I suggest converting this plot to either 3 panels, one for each model, or 4 panels, one for each transect. Currently, it is impossible to decipher.

**References**

I. J. Hewitt and C. Schoof. Models for polythermal ice sheets and glaciers. *Cryosphere*, 11 (1):541–551, 2017. doi: 10.5194/tc-11-541-2017.

N. Reeh, D. A. Fisher, R. M. Koerner, and H. B. Clausen. An empirical firn-densification model comprising ice lenses. *Ann. Glaciol.*, 42:101–106, 2005. doi: 10.3189/172756405781812871.

C. M. Stevens. *Investigations of physical processes in polar firn through modeling and field measurements.* PhD thesis, Earth and Space Sciences, 2018.

---

## Referee Comment (RC2) · Baptiste Vandecrux (Referee) · 21 Nov 2019

**Review of:**

**Advection Impacts the Firn Structure of Greenland's Percolation Zone**
**by Leone et al.**

B. Vandecrux bav@geus.dk

The presented manuscript studies the impact of ice-flow-driven displacement of snow and firn along a flowline (thereafter called horizontal advection) on the firn density, temperature, pore space and stratigraphy on the Greenland ice sheet. The authors present a novel model that includes the firn horizontal advection and analyze its sensitivity to climate forcing and meltwater routing schemes. The model is then used on four transects in the Greenland ice sheet percolation area and its output is compared with the output from a 1D model to isolate the impact of advection on the simulated firn characteristics at these locations. Eventually, the model is applied on a flowline upstream of Crawford Point, location where a firn core is available, and the impact of horizontal advection on the distribution of Melt Feature Percent through depth, a common indicator for melt, is discussed.

The presented work, and in general the study of the processes controlling the firn characteristics, are highly relevant research topics. Indeed, in a warming climate and with increasing surface melt on the Greenland ice sheet, the firn can buffer the ice sheet's sea-level contribution though the retention of meltwater. Additionally, firn characteristics such as temperature, density and stratigraphy are commonly used to describe the recent evolution of the climate. It is the first time, to my knowledge, that the firn horizontal advection is explicitly included in a firn model and that its impacts on the firn physical characteristics are being discussed. I am therefore confident that the presented manuscript has good potential for being published in the Cryosphere. However, several major limitations need to be addressed, or discussed, before publication. The major comments are listed below while specific remarks are enclosed in the commented manuscript and supplements.

1. Scope: The current manuscript does not assess where advection may have an impact on the density, temperature and stratigraphy of the Greenland firn. I believe that, from the presented model runs, simple rules based on surface velocity, topography, temperature and/or accumulation can be established to map areas where advection is relevant. Without a clear understanding of where this study applies, I am afraid that the manuscript only present model outputs and sensitivity and do not reach a sufficient scientific impact for publication in the Cryosphere.

2. Presentation of the results: In a similar way, the manuscript currently gives examples of transects where the firn advection may or may not have a significant impact on the simulated firn characteristics. These impacts are presented in a qualitative manner (in particular in the abstract and conclusion). I believe that presenting the same results in a more quantitative way would help to fully qualify the study for publication in the Cryosphere. The model can be used to give numerous metrics to quantify the impact of advection at each transect. Quantitative knowledge at each site can then be extrapolated over broader regions of the ice sheet using the rules mentioned in the previous paragraphs.

3. The structure of the manuscript should be modified because i) key information from the model are missing from the main text and should be brought in from the supplementary material: and ii) currently results are being discussed (intercompared) in the Results section and many items in the discussion are repetition from the results. The second point could be solved by either merging results and discussion sections or by clarifying what is simple description of the results and what is discussion of the results.

[revised manuscript text omitted]

---

## Author Comment (AC1) · 7 Jan 2020

**Response to Reviewer Comments**

**RC1:**
*Remarks:*
1) Improve description of modeling methods
2) Improve presentation of model results

--- Accommodate these general remarks via explicit treatment of specific comments ---

*Specific Comments:*
1. line 13: There are many missing hyphens between words, such as 'one dimensional' should be 'one-dimensional'.
Fixed.

2. line 14: I suggest 'compaction' instead of 'burial'.
Changed.

3. line 38: I think that it is worth stating that these percentages are area and not volume.
Done.

4. line 68: the two-dimensional extension deserves much more explanation!
We have expanded the paper methods to include a more robust description of our horizontal ice flow implementation.

5. line 71: I suggest adding a paragraph describing the road map for the paper.
The final paragraph on the introduction, starting line 66, has been expanded to provide a clear statement of purpose and objectives.

6. line 77: could add 'as well as compaction and advection', given that this is the topic of the paper and these clearly influence the density and thermal structure. This would strengthen the topic sentence.
We have rewritten the first two sentences of this section. However, we have kept the advection discussion in the second sentence because we see this as a modifier on the primary physics of densification.

7. line 84: 'rather' would be a good addition after the comma.
This section has been rewritten.

8. line 91: It makes sense that the one-dimension advection does a good job approximating the two-dimensional solution because the downstream gradients are small (viz. Hewitt and Schoof, 2017). The important issue, however, is that the one-dimensional advection does not include downstream transport of moisture, which is likely to be very important and is not included in the two-dimensional implementation described in the supplement, correct?

This is true. This manuscript is focused exclusively on the role of horizontal ice flow, impacts the full percolation zone. Horizontal routing of meltwater is an outstanding topic of research for the field and is beyond the scope of this work.

9. line 97: this paragraph is confusing as it makes it sound like the authors implementedeverything in FEniCS — is this true?
Yes, the model physics are simulated within the finite element package FEniCS.

10. line 120: why wasn't the Community Firn Model (Stevens, 2018) used? It contains several implementations of meltwater percolation through firn and is open source(https://github.com/UWGlaciology/CommunityFirnModel).
The model of Stevens (2018) does not include horizontal advection of the firn column. We therefore developed our own code to include horizontal transport from ice flow. As described in the text, we tested our model against other models based on the FirnMICE experiments and found favorable agreement.

11. section 2.3: I think it would be very valuable to write to out the equations in a general way, so that it is clear (1) what advance the authors have made and (2) how the advance is implemented operationally. In other words, I suggest writing the density ρ evolution equation as dρdt=f(ρ,T,...), (1)where the right-hand side is given by Reeh et al. (2005) etc. is a function of the temperature T. Then, the authors could state that they will add downstream ice advection u as ∂ρ∂t+u∂ρ∂x=f(ρ,T,...),(2)and state that this advection process occurs explicitly (my assumption), where the upstream density is advected downstream at each time step.
In our expansion of the paper's methods, we have included general equations for the densification rate of change and heat transport. These are foundational to our model implementation, which we now describe in more detail in section 2.14.

12. section 2.3.1: I suggest including some of the figures from the supplement in the main text to demonstrate how the models work.
We have moved much of the supplemental text to the manuscript body to better communicate the model mechanics. Regarding the results of the synthetic experiments, we have edited Figure 2 to show the influence of advection on simulated air content because this illustrates the primary process with which the papers is concerned. Including plots from all sensitivity tests in the manuscript body would also oblige us to describe the results from each in detail, with negative consequences for the clarity of the manuscript. We therefore have distilled these results to their most important points in the text, and leave figures of all sensitivity tests in the supplemental for interested readers.

13. line 276: 'for much another reason' should be 'for another reason'.
Fixed.

14. line 282: typo as there should be parentheses around the figure reference, i.e. '(Figure5b)'.
Fixed.

15. line 302: 'x' should be 'x'.
Fixed.

16. line 335: simulations produce data. How do I access the simulations? Also, I suggest putting Leone in parentheses to match the other funding acknowledgments.
This section has been edited and expanded to address these issues.

17. line 378: error in title
Fixed.

18. figure 1: is this figure useful/insightful?
We believe that is important to demonstrate to the reader the regions of the ice sheet which have relevance to the findings of this paper: this figure shows where high melt in the accumulation zone is coincident with relatively high horizontal displacement. In addition, this figure shows the locations of our study transects.

19. figure 2: I suggest addressing the dot, solid, and dot-dash lines within the caption. Also, an additional figure showing the two-dimensional results for the different models shown in figure 2 could be very useful (i.e. figure 3 but for the figure 2 simulations).
We have addressed the dot, solid, and dot-dash lines in the figure caption. A two-dimensional example from a sensitivity test would be useful only inasmuch as it would clarify how the sensitivity tests are performed (the results are best summarized in the figures that are already included). We have therefore improved description of the sensitivity testing in the revised manuscript methods.

20. figure 3: it would be helpful to label the subfigures on the actual figure. For example, the transect name could be put next to the letter, i.e. '(a) EGIG line' and topography, pore close-off depth, and differences could be labeled on the respective panels.
We have edited the figure panel labeling following these suggestions.

21. figure 4: I suggest converting this plot to either 3 panels, one for each model, or 4panels, one for each transect. Currently, it is impossible to decipher.
We have broken the figure in to four panels (one for each transect) to better facilitate comparison of the different infiltration schemes.

---

## Author Comment (AC2) · 7 Jan 2020

**Response to Reviewer Comments**

**RC2:**
*Major Comments:*
1. Scope: upscale model results across the ice sheet via simple 'rules'

2. Presentation of results: make results more 'quantitative' (ie. metric-based) to facilitate upscaling via major comment #1.

We address these two comments together since they are related. The purpose of this single manuscript is to demonstrate to the community that horizontal advection is an important process in determining firn structure and heat content which should not necessarily be overlooked when considering the percolation zone. At present, there is little guidance from existing literature as to whether or not this is an important process in Greenland's percolation zone.

We agree with the reviewer that analyses should be upscaled where such an exercise can be performed relatively easily while maintaining fidelity. Indeed, this is the case when considering the influence of advection on MFP, and so we have presented the methods to do so (see Discussion section 4.2). We have chosen to present methodology, rather than a quantitative map of advection-influenced MFP over the ice sheet, because presenting the methods allows interested readers to apply the methods with their own data and assumptions (e.g. averaging timescale of climate conditions) as they see fit. We view this as a much more flexible approach to upscaling for the community.

Regarding the generation of simple metrics for upscaling advection impacts on density and temperature, we believe that attempting such an exercise would be flawed for several reasons: 1) the results are sensitive functions of heat transport and densification processes which cannot simply be transformed to a common metric, 2) as we show in the manuscript, results are heavily impacted by choice of meltwater infiltration scheme which remains an outstanding challenge to the field, and 3) the results are subject to ice flow and climate variables that are prone to change in time. For these reasons, we believe that any proposed metric to simply quantify the role of advection would have limited fidelity. Without full upscale to the ice sheet, we maintain that the paper has merit because as far as we know, no existing literature has addressed this issue.

3. Modify manuscript structure: a) bring supplemental model description in to the methods, b) restrict results section to results only -- no intercomparison of results.

Our intent in the original submission was to treat the modeling methods at a high level because the objective of the paper is not to present a 'new firn model' to the community, but rather focus on the process of advection as it impacts firn structure. However, we recognize that as presented, the modeling methods may have lacked clarity for the reader without consulting the supplemental information, with negative consequences for the manuscript readability. We have therefore moved much of the supplemental model description to the methods section of the

revised manuscript, to more completely describe our numerical implementation of horizontal ice flow.

With regards to the intercomparison of results in the results section, while we understand this reasoning, the results of our methods are the comparison between two modeling scenarios. In other words, the paper is about investigating whether horizontal advection matters or not in the percolation zone, since all prior work has neglected this process. Thus, we compare 1D to 2D simulation output. We discuss the implications of the results, and with respect to prior work, in the discussion section.

*Specific Comments:*
Line 13:  Although I know now what the paper is about, there is nothing in these two sentences that explicitly define what "horizontal transport", "burial" or "advection" apply to. It would read the same if it applied on temperature advection, water vapour advection or mass advection. Consider rephrasing. Consider adding a sentence framing the study, giving its relevance on a larger scale.
We have rephrased the abstract's introductory sentence to clarify that horizontal advection acts on the entire firn column. This opening sentence places the subject in a broad context which we feel is suitable for the abstract. The introduction expands on the broader relevance of the percolation zone.

Line 17: See my general comments. These qualitative assessment are rather weak conclusions. The model, however, allows for the first time to quantify this phenomenon. So please be quantitative in the description of the impacts of advection.
Please see response to general comments above.

Line 24: I do not see the specific part in the study that support this conlcusion.
This is a discussion point; not an explicit result of the study.

Line 41: Please be more specific about how the percolation zone is related to the accumulation zone. How do you define the percolation zone?
The percolation zone was originally defined by Benson (1960); our use here is consistent with that and we do not believe repeating this information here is an efficient use of journal space. However, a related issue may be the limits we set for our modeling domains. This is more appropriately described later in the paper and is clearly illustrated in Figure 1 and its caption.

Line 52: unclear what they are
We recognize that this phrasing implies a distinction between 'evolutionary' processes and those that are not evolutionary, which is confusing. We have rephrased this sentence to eliminate this confusion.

Line 53: unclear what is the framework of the firn column

We have edited this sentence and now state explicitly that we are referring to the firn column density and ice structure.

Line 56: Please quantify.
--- this will change depending on where you are, as the paper illustrates. so quantifying is of little value and
Compared to ice divides, the percolation zone is a region of the accumulation area with high horizontal motion compared to submergence rate. --- Velocities in the perc zone can range from 20 m/yr to upwards of 1000 m/yr in areas of SE GrIS, where the accumulation zone extends nearly to the ice sheet margin.---

Line 57: Please define submergence rate
Accumulation areas of glaciers have submergent (downward) flow, whereas ablation zones have emergent (upward) flow. These are common glaciological terms (e.g., see Paterson, *Physics of Glaciers*, or Hooke, *Principals of Glacier Mechanics*).

Line 57: This is your definition of advection. Please make it clear. The word "advection" in many ways (e.g. vertical advection to describe the burial of firn under new snow).
Changed to 'horizontal advection' throughout to be more precise.

Line 62: Consider just "firn column"
Sentence reworded.

Line 63: Deep pore space do not absorb heat. Please rephrase.
We have clarified that the meltwater stored in deep pore space is a potential source of latent heat.

Line 73: Since I read the Supplementary Material after the main manuscript, I realized quite late that many crucial information were located there. Even more worrying, there are part of the model design that I completely misunderstood while reading the main text. Please bring all the model description currently in the supplement into the main text to avoid that. I leave my comments below to show which info I was looking for and which misunderstanding were created.
We have restructured and expanded the Methods in response to reviewer comments. The model physics and experiments are now more fully explained in the manuscript.

Line 75: This explanation would be very useful in the introduction.
Fair point, but we believe this is a better place so that (a) the introduction stays focused on the bigger picture, and (b) our methods have better context.

Line 83: Repeating the introduction. Consider removing. No need to describe what you do not do.
We have removed this repetitive statement from the revised manuscript.

Line 87: model?
This statement has been removed in the revised methods.

Line 94: Please move this part to the end of your model description. The reader has not seen yet what the model is about that the limitations of the model are presented. This is confusing.
We have restructured and expanded the methods section in response to the reviewer's comments. The description of model testing against the explicit 2D framework now follows the description of the physics upon which the model is based.

Line 98: please give the values/equations you use for firn thermal conductivity, heat capacity
We present the equation used for thermal conductivity and include the heat capacity value used in the revised manuscript.

Line 100: Are you tuning parameters in that schemes? If yes how? If not, which values are you using for the constants in Herron and Langway's scheme?
The Herron and Langway scheme is now presented in the Methods, clarifying the lack of tunable parameters necessary.

Line 103: give link to description
We have provided reference to the FEniCS platform in the revised methods.

Line 103: give reference
Galerkin's method is a standard solution method when using finite elements and, in our opinion, does not necessitate referencing.

Line 104: Please describe how you relate vertical velocity to accumulation rate. Which density do you use for fresh snow accumulating at the surface? Where do you get accumulation rates from? On which time scale is your model run?
These comments have been addressed in the revised methods.

Line 108: Please move to the discussion as a potential development/limitation to the study. Methods are about what was done, not what was not done.
We have opted to keep this statement in the revised methods because the outstanding challenges associated with modeling meltwater infiltration motivate our experimentation with 3 different melt schemes.

Line 112: Independently from the temperature and available pore space within the annual layer?
Yes. Reeh's method assumes the annual layer is made up of a firn fraction and ice fraction from melt without consideration of pore space or temperature.

Line 113: Although used in the cited works, the words "tipping bucket" are not appropriate to describe the model being used: In the "bucket type" water routing scheme, each layer has a storage capacity (or "bucket") defined as a fraction of the pore space (the irreducible water

content); when the "bucket" is filled, the excess water goes to the next layer but the "bucket" remains "full" and holds the water it contains; the "bucket" does not tip (which would mean empty itself in the following layer).

This model design originates from the historical concept of bucket type reservoir in hydrology. The word "tipping" was introduced by confusion with the "tipping bucket" rain gauges.

Consider removing "tipping" in the whole manuscript.

We have simplified 'tipping bucket' to simply the bucket scheme throughout the manuscript.

Line 115: please give the irreducible water content being used.
We have included the irreducible water content in the revised methods.

Line 116: larger?
We have adjusted the language to state that water percolates until the available water fails to exceed the irreducible water content.

Line 130: What is the dimension of your model domain?
We have moved this relevant information from the Supplemental to the Methods.

Line 133: Please give these ranges in a table.
We now include the ranges of values in the manuscript text.

Line 135: Please point at the relevant section of the supplementary material.
We have made this edit in the revised manuscript.

Line 136: Please give the investigated gradients.
We now list these gradients in the methods.

Line 144: How do you deal with along-transect velocity variations and mass conservation? Do you take into account material accumulating in areas of slow-down and dynamic thinning in areas of increasing velocity?
The reviewer brings up an insightful point regarding along-flow velocity variations which may influence the densification and burial rates. In short, we do not account for along-transect velocity changes in our modeling. In reality, along-flow velocity changes may influence densification rates in two ways: 1) Velocity variations may result in longitudinal deviatoric stresses that increase the effective stress, and therefore strain and densification rates within the firn profile; 2) Horizontal strain from velocity variations change the rate at which mass is added to the firn column.

Regarding mechanism 1, this effect has been found to be substantial in the special case of Antarctic ice streams (Alley, 1988), but we expect this enhanced softening to be negligibly important in Greenland's percolation zone for two reasons: 1) firn temperatures in the

percolation zone are relatively warm (cf. Antarctica) as a result of meltwater infiltration and refreezing. The warmer temperatures will decrease the firn viscosity, thereby limiting longitudinal deviatoric stresses. 2) Longitudinal deviatoric stresses go as the 2nd derivative of the velocity. Over the vast majority of Greenland's percolation zone, small velocity variations result in 2nd derivatives that are essentially negligible. As one example, Meierbachtol et al. (2016) found longitudinal resistive stresses to be very small above the long term ELA around the K-Transect.

Regarding mechanism 2, the Herron and Langway assumptions upon which our model is based, assume that the densification rate is a function of the current density, and the rate at which mass (overburden) is added (this is the accumulation rate). This is given as: $\frac{d\rho}{dt} = C(\rho_i - \rho)b\rho_i$ (note that in Herron and Langway the constant C absorbs ice density so the eqn becomes $\frac{d\rho}{dt} = k(\rho_i - \rho)$, eg. Reeh (2005)). The addition or removal of mass from ice flow effectively acts to modify the mass addition from accumulation. This magnitude of the of this mass gain/loss depends on the depth in the firn column. Deeper in the column, the mass gain/loss is amplified because the horizontal straining is acting over a larger firn thickness. The addition of a strain thinning/thickening term can be included in the equation for densification rate as: $\frac{d\rho}{dt} = C(\rho_i - \rho)\left(b\rho_i - \dot{\varepsilon}_{xx}\int_{sfc}^{z}\rho(z)dz\right)$. For clarity, we simplify the integral so that the equation reads: $\frac{d\rho}{dt} = C(\rho_i - \rho)\left(b\rho_i - \dot{\varepsilon}_{xx}z\overline{\rho(z)}\right)$. The horizontal strain is positive in stretching, and the variable z refers to the depth below the surface.

The above equation provides a convenient way to think about the influence of velocity variations on densification. Over the vast majority of the ice sheet, accumulation rates are on the order $10^{-1}$-$10^{0}$ m a$^{-1}$ (ice equivalent). In contrast, strain rates over the vast majority of the GrIS percolation zone slowly increase towards the ice margin. Horizontal strain rates are on the order $10^{-3}$ - $10^{-4}$ a$^{-1}$ (for instance, EGIG and K-transect flowlines show velocities which increase by <50 m a$^{-1}$ over the *lowest* 50 km; a strain rate of <$10^{-3}$). So when integrated over 10s of m of firn, the thinning rates are unlikely to exceed ~$10^{-2}$ m a$^{-1}$: a small fraction of accumulation.

While the above argument holds for the vast majority of the ice sheet, there are rare regions where large speed-ups along the flowline result in substantial longitudinal stretching rates. The Jakobshavn transect is one such location. Horizontal strain rates are on the order ~$10^{-2}$ in the lowest reaches of the transect. For a firn thickness of ~30 m (Figure 3) with average density ~600 m kg$^{-3}$, the resulting mass loss from thinning offset as much as ~40% of accumulation. However, in these rare locations, crevassing is likely to take up the majority of the strain, minimizing the continuous mass loss from strain thinning. This is indeed the case along Jakobshavn and even EGIG, where extensive crevassing has occurred in recent years, limiting the mobility of field parties (including ours).

We have included a sentence in the methods acknowledging this limitation of the model, and have added the above analysis to the supplemental information discussing the consequences of omitting along-flow velocity fluctuations on densification rates.

Line 146: why roughly?
Word removed.

Line 148: How do you define the number of 1D runs? Could you replace this range by a unique number of 1D run per km of transect?
We have clarified the language in the methods to state more clearly that the 1D runs were performed at annual displacements, based on the surface velocity.

Line 149: along each profile?
We have removed this confusing language.

Line 149: on?
See comment above (Line 148).

Line 150: how do you calculate the spacing from displacement?
See comment above (Line 149).

Line 157: give coordinates, average temperature, accumulation and melt rate.
We have included site location information and melt history from satellite.

Line 158: This is trivial. Consider removing. Please justify why taking this specific site as the upper limit of the percolation zone?

I understand that you need to define your transects and that it sometimes can be arbitrary. A possibility would be to state that, to limit model domain and computation time you limit your transects at the locations where melt goes below X or accumulation is above Y, which correspond to Crawford Point.

Crawford Point should however not be presented as the upper limit of percolation zone.

We have included a sentence justifying the choice of Crawford Point as a test site for investigating the role of ice flow on melt feature interpretation. It is selected because a deep core has been collected here (one of the few in Greenland's percolation zone); not exclusively because we believe Crawford Point is the upper limit of the percolation zone.

Regarding our transect modeling, we recognize that definition of the percolation zone inland extent is quite subjective, and potentially prone to large fluctuation over time. The reviewer provides no argument as to why Crawford Point should not be selected as the inland boundary. Our definition is grounded in physical process and measurement.

Line 162: By "leading to" I understand "from higher elevation down to Crawford Point"? Is it the case or is this one of the transects mentioned above? Why not presenting it a fifth transect?

What is the upper reach of this transect?

Because the objectives of this modeling exercise are unique, compared to the other 4 transects, we have opted to keep this section, rather than include this model experiment as a fifth transect. We have edited language to more explicitly state the model domain.

Line 164: what do you convert from space to time?

We have removed this statement for clarity.

Line 164: mean values over which period?

We have addressed this in revisions.

Line 166: Does that mean that you make a 100-year long run? Or do that mean that you get the spatial gradient from the 100-year average (give start/end years) temperature, melt and accumulation in RACMO for each transect?

I recommend to calculate the spatial gradient the following way for each transect: For each year, calculate the annual spatial gradient, then calculate the 100-year mean of that gradient. This should give you the same value as when calculating the spatial gradient from 100-year average forcing fields. However, with the recommended method you will be able to calculate the standard deviation that applies on the 100-years spatial gradient. It will illustrate how variable this spatial gradient was around its long-term mean value and whether the assumption of constant gradient is valid.

We have clarified that the century time scale is based on the required simulation time.

The proposed method for calculating spatial gradients essentially quantifies the interannual variability in climate gradients over the transect, whereas our interest is in long term averages. With this in mind, we have retained our existing method.

Line 167: This should be discussed properly in the Discussion section. It is important to have an idea about whether the spatial patterns in temperature, melt and accumulation are consistent enough so that firn advection would have the same effect every year.

Line 171: Please detail this, how observational studies "employed" shallow infiltration?

We have clarified that past MFP studies assume melt does not bypass the annual layer.

Line 189: Please replace by "meltwater routing scheme"

Done.

Line 202: Please give a graph of the surface altitude and climatic forcing for each transect. The reader needs to understand that the climatic gradient is not homogeneously distributed but more pronounced at lower elevation due to steeper topography.

We have included a reference to this figure, which is in the Supplemental Information.

Line 214: Please be specific: how local slope changes or even reverts the impacts of advection?

We have expanded this point with specific examples of how changes in surface slope enhance and mute the impacts of horizontal advection along the EGIG and Jakobshavn transects.

Line 221: give in %
We have quantified the reduction in surface speed relative to the Jakobshavn and EGIG transects.

Line 224: increased? decreased? be specific
Done.

Line 226: Please find a more explicit section name. Maybe "role of advection in the observed stratigraphy at Crawford Point" (also in the Methods section)
Done.

Line 227: Please be specific: which information in Figure 5 indicate the following?
We have restructured the beginning sentences of this paragraph to closely follow the figure panels in order to more explicitly walk the reader through these results.

Line 227: In this section (and in Figure 5) I find the association of depth and time not clearly defined and rather confusing. In the presence of advection, it is natural to see a gradient of MFP through depth. But it does not mean that, at that location, there was any temporal change of melt.
I recommend only discussing the MFP gradient through depth: remind that in constant climate, a 1D model gives constant MFP through depth, your model with advection gives a certain MFP gradient through depth and the observations give a third value of MFP gradient through depth.

You can then conclude that advection alone cannot explain the observed MFP gradient, confirming that the recent increase of surface melt is responsible for it.
We recognize that switching between depth- and time-gradients in MFP is somewhat non-intuitive and can generate confusion. We edited this section to more clearly explain the results in terms of changes versus depth. However, MFP has historically been used as a metric to determine time changes in climate from dated firn cores, so we also feel it is important that we present our results with respect to time. We have edited language to clearly state this conversion in the results.

Line 237: Please discuss the numerical diffusion caused by the horizontal and vertical shift of material through a fixed grid.

I am also missing a discussion point comparing the EGIG and Jakobshavn transects: they are located in the same region but, due to their differing topography, advection does not have the same impact.

Please also discuss the fact that densfication schemes have been tuned to match observed density profiles while neglecting advection. So these scheme might implicitly include the impact

of advection when used by a 1D model. The following questions can then be investigated: Which sites have been used to tune the densification scheme? Were they subject to advection?

1) Our modeling scheme is lagrangian in the horizontal dimension, eliminating the introduction of horizontal diffusion where velocities are highest. Vertical velocities are low, on the order of 1 m a$^{-1}$, minimizing the potential for numerical diffusion (as exemplified by the number of existing firn densification models in the literature which do not treat numerical diffusion).

2) We have added a section to the Discussion focused on spatial variability of firn structure that is introduced by changing ice flow. We discuss the 'local' differences between EGIG and Jakobshavn simulations, as well as the regional differences between our simulated transects.

3) Regarding the densification schemes, Herron and Langway's empirical parameterizations were determined based on a number of cores drilled at sites in Greenland and Antarctica's dry snow zone, out of the percolation zone and in a region of the ice sheet where surface speeds are small. As one example, the 'Milcent' site presented used by Herron and Langway (1980) is located ~115km inland of Crawford Point. We acknowledge that the densification parameters we employ are for dry snow zone densification in the manuscript Methods, but we view focusing discussion on the influence of advection at these dry snow sites as a non-sequitor.

Line 251: what does "it" refer to here?
Reworded.

Line 264: It is rather late in the manuscript to introduce the core. This should be moved to the methods.
Our rational for including this information in the discussion is that we are interpreting our results with respect to the findings of prior literature. Further, details about the Crawford core is not appropriate for our methods section because we did not do the work, but instead appear here with appropriate referencing.

Line 271: Do you have a reference for that?
Do you mean at Crawford Point or at the locations where the 100-year-old firn in the core was deposited?
This sentence is a natural continuation of the previous, which references Higgins (2012). Referencing Higgins again would be redundant and unnecessary.

Line 276: Please rephrase
Done.

Line 280: Is that a result from Higgins and Porter? Please be specific: how much increase over which period and with which level of significance?
We have included reference to Higgins in this sentence. The melt increase over the 1900-2007 period is reported in the previous paragraph and would therefore be redundant to repeat.

Line 281: Here it is one of your main result: be specific, how much?
We interpret this comment to largely be a matter of personal writing style. In our view, reporting a specific fraction does not enhance the impact of the result. Our point is that ice flow can generate an MFP signal that is significant and should be considered when researchers interpret MFP measured in firn cores. Whether the specific value in our example is 31% or 34% has little relevance.

Line 286: Please point at the specific section/plot where thius result has been presented.
We have edited the text to directly refer to the inland portion of the EGIG transect.

Line 292: Consider using percentage point when describing the increase of a value given in %. https://en.wikipedia.org/wiki/Percentage_point

Be specific: what fraction?
Fixed.

Line 293: This is one of the main limitation of the paper: you need to provide an estimation of where these might happen on the ice sheet.
See response to Major comments 1 and 2 above.

Line 302: Please remove "package".
Changed to parcel, because it is important to clarify flow within a 2D firn domain.

Line 302: Please reformulate this equation and distinguish x_0 the location where the firn is generated and x(t) the location where the firn is located at time t.
The purpose of this section is to provide readers with a framework for estimating the influence of horizontal advection on MFP measured in a percolation zone core. Thus, the reference location x_0 must be the core site, not the location where the firn was generated. This way, with a dated core, a user could use equation 1 to determine the location where each dated firn parcel was deposited, and compute the MFP at that location from equation 2. We have therefore kept our formulation.

Line 304: This equation is somehow already part of the model: it determines how is calculated the new position of firn from a given velocity map. This should be introduced properly in the methods: how do you shift mass in adjacent cells based on the velocity map.
The model calculates firn conditions in a time-forward sense, but here we are interested in working backwards from a core site location (not forwards from the surface to generate core conditions with depth).

The reviewer has criticized the paper for a lack of upscaling the results to the broader GrIS-scale. Yet this section is precisely where we provide this upscaling. Not every researcher has access to or interest in modeling the 2D firn conditions leading to core sites. So long as the core is dated, full modeling of the coupled densification and heat transport is not necessary. We provide a framework for any interested researcher to calculate this themselves at any point of

interest. It is quite simple to apply, which we view as an attractive feature, rather than a repetitive rephrasing of our model as the reviewer states below. Further, it is more appropriate than providing a map of advection-controlled MFP at all depths and locations across GrIS' percolation zone because it allows researchers themselves the flexibility to apply the method in ways they deem appropriate for their site/application.

We therefore have kept this section in Discussion as it is a key outcome of the paper.

Line 304: This is the definition of MFP. Needs to be in the methods.
See response above.

Line 306: In your figure 5, you already show that a 2D model including firn advection, forced with constant climate could give a depth gradient in MFP that could be misinterpreted as a temporal trend when it is in fact only the signature of spatial gradients in surface melt and advection. This paragraph and Eq.1&2 just give a mathematical rephrasing of that phenomenon and seem repetitive. Also Eq.1 and 2 are not used for anything. Consider removing the whole paragraph.
See response above.

Line 314: Please be specific: give the results for each transect.
Our objective in the conclusions is to provide a summary of results, rather than explicitly repeat the details of the results findings. With this in mind, and considering that the purpose of the manuscript is to demonstrate the process of advection (as opposed to a specific site study), we have decided to leave this statement.

Line 315: These qualitative conclusions could be made from a conceptual understanding of firn advection. With your 2D model you have the unprecedented opportunity to put numbers on these impacts. So please give numbers.
These qualitative conclusions are fundamental to understanding how advection influences firn structure. As with all models, our simulations are simplifications of the full and complex suite of processes interacting to set the firn structure across the GrIS. We therefore view the strict quantification of impacts to firn structure from advection as a fruitless exercise, and one which detracts from the main points of the manuscript. Our discussion regarding the uncertainty due to infiltration processes exemplifies why such an exercise cannot be carried out with confidence.

Line 319: I understood that the largest impacts of advection were located in over steeper regions of the percolation area. Do you expect these regions to migrate?
How the geometry of the ice sheet will evolve in the future is a very interesting question, but discussion and speculation on this topic is beyond the scope of this paper.

Line 474: red?
Fixed

Line 476: gray?
Fixed

Line 478: Am I supposed to see E, K and H letters in the figure ?
We have included letters for each transect in the revised figure.

Figure 3: add "in firn density" in the label
We referenced density in the subplot panel following suggestions from Reviewer 1.

Figure 3: add "in firn temperature" to the axis label
See above.

Figure 4: This figure is very hard to read. Please use separate panels for each transect.
Fixed

Figure 5: Although nothing is wrong in this figure, it took me a long time to understand it. Consider presenting Depth and year as y axis with 0 at the top and MFP as x axis. This will highlight that you are looking at the MFP distribution through depth at a fixed location and for a constant climate.
The intent of the figure is to communicate the influence of ice flow on interpretations of climate history using MFP, the existing literature of which has plotted time (and/or depth) on the x-axis (e.g. Graeter et al, 2018, Trusel et al., 2018, Higgins, 2012). We therefore have kept the figure as it is, which also facilitates the presentation of statistics in the manuscript in units of percentage points per year.

Line 582: Add "at Crawford Point"
Fixed

**Supplemental**

S1.3: This is not at all what I understood from the main text: you mention "we do this [running the model] over a 2D domain accounting for advective displacement" which I understand as "over a grid resolving depth and horizontal distance along a flowline". This is very different from taking a 1D model and apply a temporal gradient in surface forcing to mimic the transport of the firn column to lower elevation.
We have clarified our methods and moved corresponding supplemental material to the main manuscript body.

S1.4: Please update this equation to differentiate $z_0$, the depth to which air content is calculated and z, the variable "depth" in the integral and of which rho is a function.
This equation is accurate in its general form. The depth to which air content is calculated is a function of density of that integrated depth.

S1.4: In S6, C is presented as cumulative air content. then what is the total capacity?
Maye just add the division by rho_ice in S6 and remove the discussion of water vs. ice density.
We've add division by ice density to the equation.

S1.5: This is the only section that fits in the supplement. Please move all the rest of the supplement to the main text.
See response to general comments. Much of the supplemental is now in the manuscript body.

S2.1: Is there more processes than just ice motion? If yes, which ones? If not, just use "...the impact of ice motion on model results".
This statement is no longer in the supplemental.

S2.1: Here you mention two dimensional runs again. Is it over a grid that resolves explicitly depth and horizontal distance or is it the 1D column forced by a changing surface condition?
We have clarified this in the revised manuscript.

Figure S2: This figure should be either properly used and discussed or removed.
The figure is referenced in the manuscript text and distilled to its most important points in the presentation of the model results. For clarity, we leave the full presentation of synthetic testing in the supplemental for the most interested readers.

---

## Referee Report (RR1)

**Second review of Horizontal Ice Flow Impacts the Firn Structure of Greenland's Percolation Zone by Leone et al.**

B. Vandecrux

The new manuscript has significantly improved compared to the previous version. The model is properly presented, and the main text is now self-sufficient. I am grateful for the complete response provided by the authors which clarified some aspects of the study. Nevertheless, several rough points remain and should be clarified before it reaches the quality standard of the Cryosphere.

**General comments:**

Regarding the structure, I still believe that the splitting between results and discussion sections is confusing. For example, the two paragraphs in section 3.3. are very elusive and leave the reader wondering about the advection impact on MFP trends which only comes in the discussion section. I would strongly recommend switching to a "Results & Discussion" format, where, for each topic (Sensitivity analysis, transect comparison, MFP interpretation…), the results from the model can be presented and immediately discussed. Should the current structure be maintained, the addition of sentences guiding the reader through what is in the result and what is in the discussion would be highly beneficial.

Most of the plots lack clarity and readability:
- Y-axis in Figure 2 should be "Relative difference in air content (%)"
- Label for line colors should be added in the legends in figure 2, S2, S3.
- It took me, again, some time to interpret Figure 5b. A legend should be added to presenting each line. I also suggest making the axis label more explicit such as : "Depth in ice core" for the top axis and "Age of the firn/ice" for the bottom one.
- Figure 5a right label units should be m.
- In table 1, Why "approximate"? Check the journal's standard for abbreviations for "equiv". Also change "Speed" to "Surface velocity".  Maybe use "Surface velocity" in Figure 1 and 2 for consistency,
- In Figure 1, 2 and Table 1, three formats are used for the unit of surface velocity please use one of them and check the whole manuscript for consistency.

While the method section presented a rather complete sensitivity analysis, the description and discussion of its results appear incomplete. For instance, even though it is deemed "interesting", only three lines are dedicated to the impact of meltwater infiltration and the reader is sent to the Supplementary Material. Figure S2 should be brought to the main text and each panel should be discussed properly. The impact of Meyer and Hewitt's scheme is not presented or discussed anywhere in the manuscript. Yet it represents the most physically-based scheme and a certain "middle way" between shallow percolation and deep percolation. It should be used as a baseline when assessing the impact of advection in the transects.

**Specific comments:**

l. 130:
I understand that the model is Lagrangian with regard to the horizontal movement. Is it as well for the vertical management of layers? I remember that Meyer and Hewitt's approach was originally implemented in an Eulerian fashion: model layers have a fixed volume and each time snow is added to the top of the model, firn is shifted downward through the model layer. This approach is known to smooth firn characteristics as firn gets buried because of the repetitive averaging it implies. Could you please clarify if it is the case here?

L.191:
I am missing information about the initialization of the models. Which density, temperature, grains size profiles were used at the inland model boundary?

l.231: Please add Porter and Mosley-Thompson (2014)

l.239-240: Not sure what is meant by "increases complexity to horizontal advection signal". Please rephrase.

l.267: do you mean " the inclusion of meltwater infiltration in the firn model"? Do you have a model run where meltwater percolation is not included at all? Or do you mean compared to the "shallow percolation" scheme. Here I am also missing a presentation of the impact that the scheme of Meyer and Hewitt has on the results (see general comment).

l.274: Please replace "differs" by "decreases". Which density value is discussed here: the average for the whole column, the average above pore close off or the maximum density deviation?

l.275: replace "to pore close off" by "of pore close off"

l.311: What is the simulated pore close off depth at Crawford Point? In figure 3 it seems to be 45 m but how was it initialized? What depth was the pore close off in Higgins' core? Not sure whether it is the right place for this model validation, but this information should be somewhere to assess the model's reliability.

l.324: "unpredictable" is quite definitive, maybe fine a softer word

l.341-343: Very interesting finding. Could you specify for which variable: density, pore space, temperature or all of them? Maybe add to abstract?

l.347: replace "package" by "parcel"

l.365: This is also an interesting result. Maybe worth to be in the abstract? Please refer to figure 3 where the reader can appreciate how different the firn thickness is on the the EGIG and Jakobshavn transects.

l.390: It is unclear where does the one third value originate in Figure 5b. Could you clarify?
This part could be made more clear and impactful by referring properly to specific item in Figure 5b.

Since the contribution from advection to the MFP trend is period-dependent, it is maybe oversimplifying to give "one third" as a general contribution and it would be more useful to give the specific value for few periods:
"Over 1765-2007 period, a MFP trend of 0.08% yr-1 was found in the core compared to 0.04% yr-1 in our model that includes ice flow but no warming. This indicates that half of the trend in the core can be explained by advection. Over the 1900-2007, the observed MFP trend was 0.11% yr-1 among which X % yr-1 (please provide trend for same period) is due to advection according to our simulation. This indicates that over that period, warming contribution to MFP trend grew to X% and that the advection signal was minor."

L.395-405: This paragraph is rather confusing.
If advection is found to have a visible impact on the MFP present at Crawford Point, given that MFP is linked to firn ice content and therefore firn density, then advection should have a visible effect on firn density.

The fact that the upper EGIG transect is apparently "barely" impacted by advection is just because that impact is compared to lower areas where advection has a much higher impact.
I would therefore either remove the whole paragraph or rephrase to something such as:
"Figure 3 indicates that the firn at Crawford Point (located at the km 50 on the EGIG transect) is relatively less impacted in term of density and temperature than lower sites on the transects. Yet our results show that this relatively small impact is still sufficient to have a visible impact on MFP and on inferences regarding recent climate evolution at that site. Our work also indicates that interpretation of MFP at sites where advection has a stronger impact (lower accumulation, steeper topography or higher velocity than Crawford Point) should not be done without estimation of the advection component."

l.407-420: This paragraph seems out of place or insufficiently developed. If it aims at providing a simple way for evaluating the advection impact on MFP at a location, then it needs to be compared to ice core observations at Crawford Point and to the output of your more advanced model.
It cannot be left as "this should work". Otherwise I would recommend removing this section.

A nice addition would be to produce a map of advection-related MFP trend over the Greenland firn area. Given a 30 year period on which climate trend is usually evaluated, maps of velocity, accumulation and melt, one could use equation 5 and 6 to calculate that map. This would give a great opening to the study and definitely increase its impact.

L.429: Replace "failure" with softer word: "unsuitability", "shortcoming", "limitation"...

---

## Author Response (AR2)

**RESPONSES TO REVIEWER COMMENTS**

**Editor:**
More specifically I would advice to certainly address the comments related to the reporting of the sensitivity analysis and the quality of the plots as highlighted by reviewer 2.

> Please see changes and responses as outline below.
* * *
**Reviewer 1:**

-There is a sign error in equation (4): the advection term should be negative as written.

This correction has been made in the revised manuscript.
* * *
**Reviewer 2:**

Second review of Horizontal Ice Flow Impacts the Firn Structure of Greenland's Percolation Zone by Leone et al.

B. Vandecrux

The new manuscript has significantly improved compared to the previous version. The model is properly presented, and the main text is now self-sufficient. I am grateful for the complete response provided by the authors which clarified some aspects of the study. Nevertheless, several rough points remain and should be clarified before it reaches the quality standard of the Cryosphere.

General comments:

Regarding the structure, I still believe that the splitting between results and discussion sections is confusing. For example, the two paragraphs in section 3.3. are very elusive and leave the reader wondering about the advection impact on MFP trends which only comes in the discussion section. I would strongly recommend switching to a "Results & Discussion" format, where, for each topic (Sensitivity analysis, transect comparison, MFP interpretation...), the results from the model can be presented and immediately discussed. Should the current structure be maintained, the addition of sentences guiding the reader through what is in the result and what is in the discussion would be highly beneficial.

While we understand the argument, we disagree with this opinion. Our opinion is that if we were to combine results and discussion, this would actually make the material *more*

difficult to follow for the majority of readers; and certainly more tedious to wade through repeated 'this is a result' and 'this is an interpretation' clauses. Furthermore, this approach would orphan some material reserved for discussion sections such as our treatment of uncertainties. And, with regards to Section 3.3. we already have words to inform the reader that we will come back to interpretation of this topic in the discussion section.

Most of the plots lack clarity and readability:
-Y-axis in Figure 2 should be "Relative difference in air content (%)"

Relative difference is (difference/original) where we calculate the percent difference (difference/average). This is described in the paper's methods as such; no action taken.

-Label for line colors should be added in the legends in figure 2, S2, S3.

Understand the opinion, but we feel that it is important to minimize the clutter in the figure. In fact, many journals have explicit directions to put as much information in the caption as possible to reduce clutter in the figures. In this case, we do not believe that extracting the information from the caption is onerous.

-It took me, again, some time to interpret Figure 5b. A legend should be added to presenting each line. I also suggest making the axis label more explicit such as : "Depth in ice core" for the top axis and "Age of the firn/ice" for the bottom one.

Suggested label changes have been done. As above, we believe the merits of avoiding clutter outweigh the burden relying on the caption for this information.

-Figure 5a right label units should be m.

Fixed.

-In table 1, Why "approximate"? Check the journal's standard for abbreviations for "equiv". Also change "Speed" to "Surface velocity". Maybe use "Surface velocity" in Figure 1 and 2 for consistency,

Table fixed.
We label the figures with 'Speed' because the entity here is magnitude (velocity = magnitude + direction). Text and figures are consistent with our use of speed.

-In Figure 1, 2 and Table 1, three formats are used for the unit of surface velocity please use one of them and check the whole manuscript for consistency.

Done; consistent.

While the method section presented a rather complete sensitivity analysis, the description and discussion of its results appear incomplete. For instance, even though it is deemed "interesting", only three lines are dedicated to the impact of meltwater infiltration and the reader is sent to the Supplementary Material. Figure S2 should be brought to the main text and each panel should be discussed properly. The impact of Meyer and Hewitt's scheme is not presented or discussed anywhere in the manuscript. Yet it represents the most physically-based scheme and a certain "middle way" between shallow percolation and deep percolation. It should be used as a baseline when assessing the impact of advection in the transects.

Again, we understand the argument behind this opinion, but our opinion is that the paper must balance presentation of results with readability. Documentation of all details of the sensitivity analysis would detract from the most important points of the manuscript, and Figures 2 and 4 already present comparisons between different melt infiltration schemes for the synthetic and real world scenarios.

We do, however recognize that results presented in the figures should be discussed in the text. We have therefore added text to the results section describing the Meyer and Hewitt output from the sensitivity testing and transect modeling.

Specific comments:

l. 130:
I understand that the model is Lagrangian with regard to the horizontal movement. Is it as well for the vertical management of layers? I remember that Meyer and Hewitt's approach was originally implemented in an Eulerian fashion: model layers have a fixed volume and each time snow is added to the top of the model, firn is shifted downward through the model layer. This approach is known to smooth firn characteristics as firn gets buried because of the repetitive averaging it implies. Could you please clarify if it is the case here?
This has been reworded to clarify.

L.191:
I am missing information about the initialization of the models. Which density, temperature, grains size profiles were used at the inland model boundary?
This information has been added.

l.231: Please add Porter and Mosley-Thompson (2014)
We are confused by this request as this paper has nothing to do with ice surface velocity. No action taken.

l.239-240: Not sure what is meant by "increases complexity to horizontal advection signal". Please rephrase.
Reworded.

l.267: do you mean " the inclusion of meltwater infiltration in the firn model"? Do you have a model run where meltwater percolation is not included at all? Or do you mean compared to the "shallow percolation" scheme. Here I am also missing a presentation of the impact that the scheme of Meyer and Hewitt has on the results (see general comment).
Sentence reworded to clarify.

In addition, we have included a paragraph explicitly describing sensitivity testing results from the suite of infiltration schemes. This includes the continuum scheme from Meyer and Hewitt.

l.274: Please replace "differs" by "decreases". Which density value is discussed here: the average for the whole column, the average above pore close off or the maximum density deviation?
Done.

l.275: replace "to pore close off" by "of pore close off"
Fixed.

l.311: What is the simulated pore close off depth at Crawford Point? In figure 3 it seems to be 45 m but how was it initialized? What depth was the pore close off in Higgins' core? Not sure whether it is the right place for this model validation, but this information should be somewhere to assess the model's reliability.

Fixed. This is a misunderstanding by the reviewer, as Figure 3 only presents the lowest 50 km of each transect, and therefore does not include Crawford Point. We have changed the transect extents in Figure 3 and supplemental Figures S4 and S5 to show the full transect results.

We do indeed include a statement concerning the model validation in Section 4.3. Our model age is within 7% of that presented by Higgins.

l.324: "unpredictable" is quite definitive, maybe fine a softer word
Fixed.

l.341-343: Very interesting finding. Could you specify for which variable: density, pore space, temperature or all of them? Maybe add to abstract?
Done. And, this was already in the conclusions.

l.347: replace "package" by "parcel"
Done.

l.365: This is also an interesting result. Maybe worth to be in the abstract? Please refer to figure 3 where the reader can appreciate how different the firn thickness is on the the EGIG and Jakobshavn transects.
Indeed, the paragraph opens with a reference to Figure 3, two sentences prior. A second call out here would be repetitive and unnecessary; no action taken.

l.390: It is unclear where does the one third value originate in Figure 5b. Could you clarify? This part could be made more clear and impactful by referring properly to specific item in Figure 5b. Since the contribution from advection to the MFP trend is period-dependent, it is maybe oversimplifying to give "one third" as a general contribution and it would be more useful to give the specific value for few periods: "Over 1765-2007 period, a MFP trend of 0.08% yr-1 was found in the core compared to 0.04% yr-1 in our model that includes ice flow but no warming. This indicates that half of the trend in the core can be explained by advection. Over the 1900-2007, the observed MFP trend was 0.11% yr-1 among which X % yr-1 (please provide trend for same period) is due to advection according to our simulation. This indicates that over that period, warming contribution to MFP trend grew to X% and that the advection signal was minor."

This section has been reworded to accommodate this suggestion, and to improve the overall clarity.

L.395-405: This paragraph is rather confusing.
If advection is found to have a visible impact on the MFP present at Crawford Point, given that MFP is linked to firn ice content and therefore firn density, then advection should have a visible effect on firn density. The fact that the upper EGIG transect is apparently "barely" impacted by advection is just because that impact is compared to lower areas where advection has a much higher impact. I would therefore either remove the whole paragraph or rephrase to something such as:

"Figure 3 indicates that the firn at Crawford Point (located at the km 50 on the EGIG transect) is relatively less impacted in term of density and temperature than lower sites on the transects. Yet our results show that this relatively small impact is still sufficient to have a visible impact on MFP and on inferences regarding recent climate evolution at that site. Our work also indicates that interpretation of MFP at sites where advection has a stronger impact (lower accumulation, steeper topography or higher velocity than Crawford Point) should not be done without estimation of the advection component."

We have reworded this paragraph to clarify, including some of the wording suggested. However, we feel it is important to take the further step of explaining how this non-intuitive result can be explained. Indeed, the first sentence of this response seems to question the result. The second part of the paragraph provides discussion. We have also reworded this section for clarity.

l.407-420: This paragraph seems out of place or insufficiently developed. If it aims at providing a simple way for evaluating the advection impact on MFP at a location, then it needs to be compared to ice core observations at Crawford Point and to the output of your more advanced model. It cannot be left as "this should work". Otherwise I would recommend removing this section. A nice addition would be to produce a map of advection-related MFP trend over the Greenland firn area. Given a 30 year period on which climate trend is usually evaluated, maps of velocity, accumulation and melt, one could use equation

5 and 6 to calculate that map. This would give a great opening to the study and definitely increase its impact.

While producing such a map is technically trivial, we feel this would violate our personal standards for scientific quality. Regional climate models have substantial and non-uniform uncertainty, and the velocity datasets for much of the percolation zone, especially at higher elevations, are not up to the task. For example, velocity products for the featureless snow covered regions contain voids and artifacts; in fact, some products are actually a computed balance velocity over these areas. So our position is that using these data for such an analysis could suffer from garbage-in/garbage-out. Instead, we provide an analytical solution to the problem so that any future researcher can apply this with the opportunity to make their own assessment of data quality for their region of interest.

L.429: Replace "failure" with softer word: "unsuitability", "shortcoming", "limitation"...
Done.